# FinLoRA: Benchmarking LoRA Methods for Fine-Tuning LLMs on Financial Datasets

## Abstract

Low-rank adaptation (LoRA) methods show great potential for scaling pre-trained general-purpose Large Language Models (LLMs) to hundreds or thousands of use scenarios. However, their efficacy in high-stakes domains like finance is rarely explored, e.g., passing CFA exams and analyzing SEC filings. In this paper, we present the open-source FinLoRA project that benchmarks LoRA methods on both general and highly professional financial tasks. First, we curated 19 datasets covering diverse financial applications; in particular, we created four novel XBRL analysis datasets based on 150 SEC filings. Second, we evaluated five LoRA methods and five base LLMs. Finally, we provide extensive experimental results in terms of accuracy, F1, and BERTScore and report computational costs in terms of time and GPU memory during fine-tuning and inference stages. We find that LoRA achieved substantial performance gains of 40.1 points on average over base models. Our FinLoRA project provides an affordable and scalable approach to democratize financial intelligence to the general public. Datasets, LoRA adapters, code, and documentation are available at `https://anonymous.4open.science/r/FinLoRA-ICLR-115E`.

## 1 Introduction

Large language models (LLMs) have demonstrated impressive general capabilities in various vertical domains, such as finance (Wu et al., 2023; Liu et al., 2023a; Lee et al., 2025), education (Liu et al., 2024b), and scientific discovery (Lu et al., 2022; Chen et al., 2021). In the finance sector, LLMs have been applied to tasks such as sentiment analysis (Zhang et al., 2023), question-response, and stock market prediction (Lee et al., 2025).

Cost-effective adaptation is critical for applying LLMs to vertical domains like finance, since general-purpose LLMs lack the specialized knowledge to excel in professional-level tasks. Full fine-tuning can close such performance gaps but is prohibitive for most organizations due to its computationally demanding nature. As such, parameter-efficient fine-tuning (PEFT), particularly Low-Rank Adaptation (LoRA) (Hu et al., 2022) and its variants (Dettmers et al., 2023; Mao et al., 2024; Yang et al., 2024; Huan & Shun, 2025; Liu et al., 2022; Chen et al., 2022; Meng et al., 2024), has emerged as an affordable and scalable solution. LoRA methods can enhance pre-trained general-purpose LLMs with domain-specific knowledge and improve performance on downstream tasks (Mao et al., 2024).

Recent research like FinGPT (Liu et al., 2023a; 2024e) has applied a quantized LoRA method (Dettmers et al., 2023) to general financial tasks; however, the comparative performance of various LoRA variants in complex, professional-level financial tasks remains rarely explored. Previous research shows that LLMs are struggling with professional-level financial tasks, such as analyzing SEC filings (Islam et al., 2023) and passing financial certificate exams (Callanan et al., 2024). A critical area within professional finance involves eXtensible Business Reporting Language (XBRL) data (Saeedi et al., 2007), the de facto global standard for business reporting. Despite XBRL's importance, dedicated datasets for related analytical tasks are scarce. This deficiency, coupled with the need to evaluate different LoRA methods on highly specialized financial tasks, motivates our introduction of FinLoRA: a comprehensive benchmark designed to assess LoRA variants across diverse financial scenarios, with an emphasis on professional XBRL applications.

This paper demonstrates that fine-tuning state-of-the-art LLMs can significantly improve performance across a range of financial tasks, including specialized XBRL analysis, locally and cost-effectively

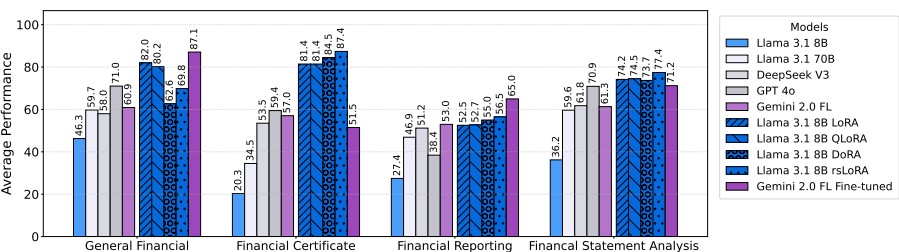

Figure 1: Average performance of base models and LoRA models.

using widely accessible GPUs. As illustrated in Fig. 1, our LoRA-adapted models achieve notable performance improvements over baseline models across four categories of financial tasks. Our main contributions are summarized as follows:

- We curated 19 financial datasets, including general financial tasks, financial analysis, and professional-level XBRL tasks. In particular, we created four novel XBRL analysis datasets. This enables future research to perform rigorous evaluation of LoRA methods in financial tasks.

- We implemented and fairly compared five LoRA methods—including LoRA (Hu et al., 2022), QLoRA (Dettmers et al., 2023), DoRA (Liu et al., 2024c), rsLoRA (Kalajdzievski, 2023), and Federated LoRA—by fine-tuning models on financial datasets. LoRA achieved an average increase of 40.1 point increase in accuracy over baseline models, which validates the effectiveness of low-rank adaptation and quantization for fine-tuning LLMs.

- We conducted an extensive analysis with 59 rounds of fine-tuning and ∼300 rounds of evaluations for LoRA methods from four angles: (i) a comprehensive comparison across different base models and datasets, (ii) performance on various types of financial tasks, (iii) resource requirements for fine-tuning and inference, and (iv) practical considerations for LoRA deployment in finance.

## 2  Is Fine-tuning of LLMs Needed on Financial Tasks?

While general-purpose LLMs demonstrate broad NLP competence, their performance often falls short on nuanced financial tasks (Islam et al., 2023; Callanan et al., 2024). This section discusses three key reasons that underscore the necessity of fine-tuning, particularly with methods like LoRA, for developing effective financial LLMs: ***(i) Lack of High-Quality Financial Data in Pre-Training Datasets.*** Many pre-training datasets, such as The Pile (Gao et al., 2020), primarily draw from general web crawls (e.g., GitHub, arXiv). These sources often under-represent high-quality, specialized financial data, which may be private and exist in complex formats (like XBRL). Consequently, to equip LLMs with the understanding required for complex financial analysis, targeted fine-tuning on curated, domain-specific datasets becomes essential. ***(ii) General LLMs' Failure in Specialized Financial Tasks.*** General LLMs often struggle with specialized tasks that demand deep domain-specific knowledge. XBRL analysis provides a clear illustration of these difficulties: Table 1 details the typical Llama base model's errors on XBRL questions and its improved outputs after LoRA fine-tuning. In Question 1, the base model hallucinated and created a non-existent tag that is not part of the US GAAP taxonomy. In Question 2, the base model did not know the formula of the equity multiplier, resulting not select the correct assets value. ***(iii) Cost and Time.*** As shown in Table 5, the training-from-scratch approach of BloombergGPT (Wu et al., 2023), which reportedly cost $2.7 million and took 53 days to train (Table 5), is economically nonviable for most organizations. In contrast, fine-tuning existing foundational models using LoRA methods is significantly more accessible and time-efficient.

## 3  FinLoRA Benchmark

### 3.1  Benchmark Tasks, Datasets, and Metrics

As displayed in Table 2, we consider four types of tasks: general financial tasks, financial certificate tasks, financial reporting tasks, and financial statement analysis tasks.

Table 1: Case study—XBRL tagging (Google 10-Q 2025-Q1) and XBRL formula calculation (Travelers 10-K FY-2023)—from base Llama 3.1 8B Instruct and our LoRA fine-tuned version.

| Question 1 | What is the appropriate XBRL US-GAAP tag for "2.0" in 3 **"...equity securities accounted for under the equity method had a carrying value of approximately $2.0 billion"**? |
|---|---|
| Llama 3.1 8B | `us-gaap:MajorityEquityInterest` |
| Llama 3.1 8B LoRA (8bit r8) | `us-gaap:EquityMethodInvestments` |
| Ground truth | `us-gaap:EquityMethodInvestments` |
| **Question 2** | What is Travelers Companies Inc's **Equity Multiplier** for FY 2023? (Answer with a formula substituted with values.) {XBRL Context} |
| Llama 3.1 8B | $(1,209,000,000 \ / \ 249,210,000,000)$ |
| Llama 3.1 8B LoRA (8bit r8) | $125,978,000,000 \ / \ 249,210,000,000$ |
| Ground truth | $125,978,000,000 \ / \ 249,210,000,000$ |

**Public Financial Datasets**   FinLoRA includes 15 public financial datasets. *(i)* Sentiment analysis (SA): Financial Phrase Bank (FPB) (Malo et al., 2013), Financial QA Sentiment Analysis (FiQA SA) (Maia et al., 2018), Twitter Financial News Sentiment (TFNS) (Rahman, 2022), and News with GPT Instruction (NWGI) (Liu et al., 2024e), each with financial text from news or tweets and sentiment labels. *(ii)* Headline analysis: The Headline dataset (Sinha & Khandait, 2020) classifies financial headlines based on various questions into two classes: "yes" and "no". *(iii)* Named-entity recognition (NER): NER dataset (Salinas Alvarado et al., 2015) annotates one entity per sentence, categorized into one of three classes: "location", "person", and "organization". *(iv)* Financial certification tasks with a focus on regulations/ethics: CFA Level I, II, and III, and CPA Regulation. *(v)* Financial reporting: XBRL Terminology (Han et al., 2024), Financial Numeric Entity Recognition (FiNER) (Loukas et al., 2022), and Financial Numeric Extreme Labeling (FNXL) (Sharma et al., 2023). *(vi)*: Financial statement analysis: Financial Math (Han et al., 2024) and FinanceBench (Islam et al., 2023; Han et al., 2024).

**Newly-added XBRL Analysis Datasets**   We introduce 4 novel XBRL analysis datasets, i.e., extracting and analyzing SEC financial reports in XBRL format. These question-answering datasets, derived from the 2019-2023 annual reports of Dow Jones 30 companies, provide each example with a question, a relevant filtered XBRL text segment as source material, and a ground truth answer. The datasets cover four distinct task types: *(i)* **XBRL tag extraction** involves extracting a specific XBRL tag from a raw XBRL text segment given a natural language description of the tag. *(ii)* **XBRL value extraction** focuses on extracting a numeric value from the raw XBRL text segment given a natural language description of the value. *(iii)* **XBRL formula construction** tasks the LLM to first identify and select multiple relevant facts (and their corresponding XBRL tags) from the XBRL data, and then construct a standard financial formula (e.g., Net Profit Margin, Quick Ratio) using these selected tags as components. *(iv)* **XBRL formula calculation** builds on the previous task and requires the LLM to substitute the actual numeric values into the formula and compute the final result.

**Dataset Construction Pipeline**   Initially, we classified financial tasks into nine categories, creating a training set for each to develop category-specific LoRA adapters per configuration. The four novel XBRL analysis datasets were constructed using XBRL-formatted 10-K annual reports from Dow Jones 30 companies (2019-2023). For these, we generated the four aforementioned types of questions by applying five distinct templates to consolidated, company-specific facts. To ensure contextual relevance, XBRL file segments were automatically filtered based on pertinent factors like year and reporting axes.

**Metrics**   For all general financial tasks, financial analysis, XBRL tagging, financial math, and XBRL analysis tasks, we use Exact Match (EM) to evaluate the LLMs' output and report both the accuracy and weighted F1 score (in the supplementary materials). For XBRL Term and FinanceBench, we report BERTScore F1 (Zhang et al., 2020) instead. We also report the average of scores across the tasks with BERTScore F1 multiplied by 100.

Table 2: Benchmark tasks and datasets.

| Datasets | Types | #Train/#Test | Average Prompt Length | Metrics | Sources & License |
|---|---|---|---|---|---|
| **General Financial Tasks** (Total: 122.9k/31.7k) | | | | | |
| FPB (Malo et al., 2013) | Sentiment Analysis | 3.1k/970 | 56 | Accuracy, F1 | HF, CC BY-SA 3.0 |
| FiQA SA (Maia et al., 2018) | Sentiment Analysis | 822/234 | 48 | Accuracy, F1 | HF MIT |
| TFNS (Rahman, 2022) | Sentiment Analysis | 9.5k/2.4k | 52 | Accuracy, F1 | HF MIT |
| NWGI (Liu et al., 2023a) | Sentiment Analysis | 12.9k/4.1k | 81 | Accuracy, F1 | HF MIT |
| Headline (Sinha & Khandait, 2020) | Headline Analysis | 82.2k/20.5k | 43 | Accuracy, F1 | HF CC BY-SA 3.0 |
| NER (Salinas Alvarado et al., 2015) | NER | 13.5k/3.5k | 138 | Accuracy, F1 | HF CC BY-SA 3.0 |
| **Financial Certificate Tasks** (Total: 472/346) | | | | | |
| CFA Level I | Analyst Exam | 180/90 | 181 | Accuracy, F1 | Internet (Public; Not Released Due to Copyright) |
| CFA Level II | Analyst Exam | 88/77 | 1.0k | Accuracy, F1 | |
| CFA Level III | Analyst Exam | 80/78 | 961 | Accuracy, F1 | |
| CPA REG | Accountant Exam | 124/101 | 147 | Accuracy, F1 | |
| **Financial Reporting Tasks** (Total: 15.9k/8.3k) | | | | | |
| FiNER-139 (Loukas et al., 2022) | XBRL Tagging | 10.0k/7.4k | 1.8k | Accuracy, F1 | HF CC BY-SA 4.0 |
| FNXL (Sharma et al., 2023) | XBRL Tagging | -/247 | 7.1k | Accuracy, F1 | GitHub Public |
| XBRL Term (Han et al., 2024) | Terminology | 5.9k/651 | 25 | BERTScore | GitHub MIT |
| **Financial Statement Analysis Tasks** (Total: 27.9k/7.3k) | | | | | |
| Financial Math (Han et al., 2024) | Math | 800/200 | 116 | Accuracy | GitHub MIT |
| FinanceBench (Islam et al., 2023; Han et al., 2024) | Math | 86/43 | 983 | BERTScore | GitHub CC BY-NC 4.0 |
| Tags Extraction | XBRL Analysis | 10.1K/2.9k | 3.8k | Accuracy, F1 | Supplementary Materials MIT |
| Values Extraction | XBRL Analysis | 10.1k/2.5k | 3.8k | Accuracy, F1 | Supplementary Materials MIT |
| Formula Construction | XBRL Analysis | 3.4K/835 | 3.8k | Accuracy, F1 | Supplementary Materials MIT |
| Formula Calculation | XBRL Analysis | 3.4K/835 | 3.8k | Accuracy, F1 | Supplementary Materials MIT |

## 3.2 BASE MODELS AND LoRA METHODS

**Base Models** We benchmark two models for both base model and LoRA fine-tuning performance—Llama 3.1 8B Instruct (Dubey et al., 2024) and Gemini 2.0 Flash Lite (Team et al., 2024). We also evaluated three additional models—Llama 3.1 70B Instruct (Dubey et al., 2024), DeepSeek V3 (Liu et al., 2024a), and GPT-4o (Hurst et al., 2024)—as base models only.

**LoRA Methods** We considered the following five popular LoRA methods.

- (Vanilla) **LoRA**: Low-rank adaptation (LoRA) (Hu et al., 2022) is a parameter-efficient fine-tuning method that preserves the weights of the pre-trained model and introduces a smaller set of trainable weights. The updated weights follow the low-rank decompositions $\Delta W = \gamma_r BA$, where $\gamma_r$ is a scaling factor ($\gamma_r = \frac{\alpha}{r}$ with $\alpha > 0$ and rank $r > 0$), $A \in \mathbb{R}^{r \times k}$ and $B \in \mathbb{R}^{d \times r}$ are trainable parameters, and $W_0 \in \mathbb{R}^{d \times k}$ denote the pre-trained weights. During the fine-tuning stage, the forward pass is $y = (W_0 + \gamma_r BA)x = W_0 x + \gamma_r BAx$.

- **QLoRA**. Quantized LoRA (QLoRA) (Dettmers et al., 2023) further reduces memory usage by using 4-bit quantization. During fine-tuning, all weights of the pre-trained model are quantized to 4 bits. Weights will be dynamically dequantized back to 16 bits when performing computation with the input sequence $x$ and the adapter matrix $A$ and $B$, which remain in 16-bit precision throughout the process, where $y = p_{16}(W_0^{\text{NF4}})x + \gamma_r BAx$. The process is similar in the inference stage, where the merged weights $W$ are loaded in 4-bit precision.

- **DoRA**. Weight-Decomposed Low-Rank Adaptation (DoRA) (Liu et al., 2024c) decomposes $W_0 \in \mathbb{R}^{d \times k}$ into a column-wise magnitude vector $m \in \mathbb{R}^{1 \times k}$ and a direction matrix $V \in \mathbb{R}^{d \times k}$, where $m = \|W_0\|_c$ (with $\|\cdot\|$ being column-wise norm) and $V = W_0$. Only the direction matrix receives updates through LoRA. The magnitude vector is updated separately. DoRA can achieve accuracy close to that from full fine-tuning while keeping the same parameter count as LoRA.

- **rsLoRA**. Vanilla LoRA uses a scaling factor $\alpha/r$, which may cause gradients to explode or diminish as the rank $r$ increases. Rank-Stabilized LoRA (rsLoRA) (Kalajdzievski, 2023) uses a scaling factor $\alpha/\sqrt{r}$: $W' = W_0 + \frac{\alpha}{\sqrt{r}} BA$. This scaling results in gradient-scale stability at higher ranks, enabling the rank to be higher for long-context tasks like XBRL analysis.

- **LoRA with Federated Learning**. In the finance sector, multiple institutions may want to collaborate using their own proprietary datasets, but they cannot share their data due to compliance reasons and privacy concerns. Federated learning solves this issue by fine-tuning a model on local data and aggregating LoRA updates to a central node.

### 3.3 BENCHMARK ANGLES

**Angle I: LoRA Methods' Performance on Financial Datasets**   We seek to learn which LoRA method is most effective in financial tasks, in terms of both category-specific and overall performance, and how these LoRA fine-tuned models perform compared to existing state-of-the-art (SOTA) models. We fine-tuned Llama 3.1 8B Instruct using LoRA, QLoRA, rsLoRA, and DoRA, representing open-source models and fine-tuning approaches, and fine-tuned Gemini 2.0 Flash Lite using Google's proprietary fine-tuning methods as a baseline representing closed-source counterparts.

**Angle II: LoRA Suitability for Financial Tasks**   We wish to investigate how the benefits of LoRA fine-tuning vary across different financial tasks. This angle is motivated by the need to identify which specific applications (e.g., sentiment analysis, XBRL tagging, XBRL analysis) are most responsive to fine-tuning, and what properties of the datasets cause this.

**Angle III: Resources of LoRA Fine-tuning and Inference**   We aim to compare which LoRA methods, out of the tested methods, are the most cost-effective in fine-tuning and compare the fine-tuning cost to closed-source fine-tuning services. We are also motivated to measure and compare the inference speeds of LoRA-fine-tuned models against their larger base model counterparts. The goal is to quantify the potential for reduced latency and increased throughput, which are critical for real-time financial applications and operational efficiency.

**Angle IV: Practical Considerations for LoRA Deployment in Finance**   To assess the viability of deploying LoRA-fine-tuned models in real-world financial scenarios, we investigate two key concerns: *(i)* Data Privacy in Collaborative Training: While local LoRA fine-tuning enhances data protection, collaborative model training across multiple institutions often requires approaches like Federated Learning to preserve the privacy of proprietary training data. We investigate this by simulating data distribution across several nodes and evaluating LoRA fine-tuning performance against centralized training. *(ii)* Catastrophic Forgetting: Fine-tuning can risk degrading a model's pre-existing general knowledge and capabilities. To quantify this, we evaluate our LoRA-fine-tuned models on established general-domain benchmarks, such as MMLU (Hendrycks et al., 2020), measuring any performance changes on tasks outside their financial fine-tuning scope.

## 4 BENCHMARK RESULTS

**Setup**   Our experiments were conducted on four NVIDIA A5000 GPUs. For closed-source models, we employed various inference and fine-tuning APIs. *(i)* **Single-task fine-tuning**: we separate our datasets into 9 tasks, for each tasks we fine-tune using vanilla LoRA, QLoRA, DoRA, rsLoRA on Llama 3.1 8B Instruct, and only vanilla LoRA on Ministral 8B Instruct, and Google proprietary tunning on Gemini 2.0 Flash Lite. *(ii)* **Multi-task fine-tuning**: We also conducted multi-task fine-tuning that only outputs 4 LoRA adapters based on the major categories. We used a learning rate of 1e-4 and a batch size of 2–8 based on prompt length (Refer to appendix section C for details).

### 4.1 ANGLE I: LORA METHODS PERFORMANCE ON FINANCIAL DATASETS

**Comparative Performance of LoRA Variants**   Table 4 shows the performance of base models and different LoRA fine-tuned models. Vanilla LoRA (8-bit, rank 8) achieves the highest overall average score (74.74), a 40.1 point increase over the Llama 3.1 8B base model. Fig. 1 shows the performance by category. Vanilla LoRA outperforms other LoRA variants in general financial tasks, while rsLoRA leads in financial analysis, financial reporting, and financial statement analysis.

**Single-task vs. Multi-task Fine-tuning**   Multi-task fine-tuning produces clear gains in Financial Statement Analysis. Formula construction, formula calculation, Finance Bench, and financial

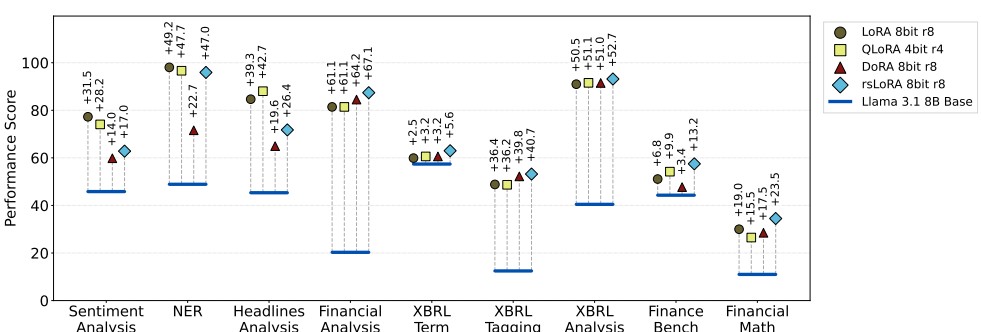

Figure 2: Task suitability.

math. These tasks share similar underlying knowledge like the structure of financial statements and numerical reasoning. Learning them together helps the model to achieve positive knowledge transfer. In contrast, we see negative transfer in General Financial and Financial Reporting tasks, where the multi-task model performs worse on TFNS, Headline, FiNER, and FNXL. The problem comes from differences in task format and objective. Even with balanced sampling, the model struggles to optimize for very different objectives at the same time. Overall, performance drops by 6 points when tasks are merged. Closely related tasks can benefit from joint training, while divergent tasks often harm each other. Refer to appendix section A.3 for more details.

**rsLoRA Performs Better at High Ranks** rsLoRA scales with $\alpha/\sqrt{r}$ instead of $\alpha/r$ to prevent gradient exploding or vanishing at large ranks. We set $r = 8$ for memory efficiency. rsLoRA just slightly underperforms against LoRA and QLoRA. The rsLoRA paper's experiments (Kalajdzievski, 2023) led to lower perplexity at higher ranks (e.g., $r = 64$). This lower perplexity and the fact that higher rank LoRA captures more details suggest rsLoRA's benefits are primarily exploited at high ranks.

**DoRA Benefits from Two Learning Rates** DoRA performed worse than the other three LoRA methods. We used the same learning rate for updating the magnitude vector and direction matrix. However, as shown in Table 4, this can lead to sub-optimal performance in some cases due to the gradient scales being different between the two types of updates in DoRA. This leads to DoRA sometimes under-training the magnitude vector in our experiments, which uses the same low learning rate. Thus, DoRA may achieve higher performance if the magnitude vector has its own learning rate that is higher than the low-rank update's learning rate.

**LoRA-Tuned Llama 3.1 8B vs. Baseline Models and Gemini Fine-Tuned** Compared to SOTA base LLMs, the LoRA-tuned Llama 3.1 8B Instruct models generally show superior performance across most datasets, with NWGI and FNXL being the exceptions. Against another fine-tuned baseline, the Gemini 2.0 FL fine-tuned model, this Gemini model excels in general financial tasks and XBRL data reporting. However, our Llama 3.1 8B Instruct LoRA variants demonstrate stronger average performance in financial analysis and XBRL data analysis tasks.

## 4.2 ANGLE II: FINANCIAL TASK LoRA SUITABILITY

Fig. 2 highlights LoRA's varying effectiveness across different financial tasks. A key observation is the contrast in LoRA method improvements between XBRL Analysis tasks and FinanceBench. Although both aim to analyze financial statements, tasks based on XBRL data demonstrate substantial LoRA-induced performance improvements, whereas FinanceBench exhibits minimal gains. This disparity underscores XBRL's superior suitability for financial statement analysis. The standardized semantics and taxonomy inherent in XBRL likely provide a more structured and consistent learning environment for LLMs, facilitating more effective adaptation compared to FinanceBench, which relies on OCR-processed PDF data lacking such rich, standardized metadata. These findings emphasize the crucial role of XBRL in enabling effective LLM integration for financial report analysis.

Table 4: Performance on financial tasks: accuracy in blue, F1 in gray, and BERTScore F1 in green.

| | Base Models | | | | | | | Fine-tuned Models | | | | | | |
|---|---|---|---|---|---|---|---|---|---|---|---|---|---|---|
| | 0 Shot | | | | | | 3 Shot | Single-task | | | | | | Multi-task |
| **Datasets** | Llama 3.1 8B | Mini-stral 8B | Gemini 2.0 FL | Llama 3.1 70B | Deep-Seek V3 | GPT-4o | Llama 3.1 8B | Llama 3.1 8B LoRA | Llama 3.1 8B QLoRA | Llama 3.1 8B DoRA | Llama 3.1 8B rsLoRA | Mini-stral 8B LoRA | Gemini 2.0 FL N/A | Llama 3.1 8B LoRA |
| **General Financial Tasks** | | | | | | | | | | | | | | |
| FPB | 68.73 | 73.08 | 81.02 | 74.50 | 78.76 | 81.13 | 76.40 | 85.64 | 84.16 | 81.93 | 82.84 | 86.71 | **87.62** | 85.31 |
| | 0.677 | 0.695 | 0.894 | 0.736 | 0.764 | 0.818 | 0.865 | **0.922** | 0.909 | 0.901 | 0.853 | 0.868 | 0.878 | 0.873 |
| FiQA SA | 46.55 | 52.86 | 68.09 | 47.27 | 60.43 | 72.34 | 64.68 | 81.28 | 78.30 | 78.72 | 73.19 | 80.00 | **88.09** | 82.20 |
| | 0.557 | 0.591 | 0.810 | 0.565 | 0.686 | 0.773 | 0.782 | **0.884** | 0.874 | 0.874 | 0.806 | 0.807 | 0.879 | 0.853 |
| TFNS | 24.16 | 22.07 | 26.38 | 68.42 | 84.38 | 73.32 | 28.81 | 88.02 | 83.84 | 59.09 | 59.51 | 45.85 | **89.49** | 34.51 |
| | 0.339 | 0.189 | 0.385 | 0.686 | 0.846 | 0.740 | 0.415 | **0.932** | 0.910 | 0.702 | 0.655 | 0.460 | 0.896 | 0.399 |
| NWGI | 43.86 | 21.25 | 48.16 | 50.14 | 7.44 | **66.61** | 32.20 | 54.16 | 49.96 | 19.57 | 35.80 | 56.90 | 62.59 | 36.51 |
| | 0.583 | 0.226 | 0.614 | 0.596 | 0.097 | 0.656 | 0.399 | **0.690** | 0.645 | 0.281 | 0.464 | 0.538 | 0.581 | 0.475 |
| NER | 48.89 | 58.61 | 65.13 | 46.28 | 40.82 | 52.11 | 58.85 | **98.05** | 96.63 | 71.59 | 95.92 | **98.05** | 97.29 | 76.07 |
| | 0.569 | 0.647 | 0.769 | 0.454 | 0.360 | 0.523 | 0.707 | **0.981** | 0.966 | 0.834 | 0.963 | **0.981** | 0.973 | 0.856 |
| Headline | 45.34 | 62.64 | 76.60 | 71.68 | 76.06 | 80.53 | 64.28 | 84.66 | 88.03 | 64.93 | 71.75 | **97.51** | 97.32 | 13.90 |
| | 0.558 | 0.660 | 0.847 | 0.729 | 0.779 | 0.814 | 0.782 | 0.852 | 0.886 | 0.781 | 0.828 | **0.976** | 0.973 | 0.197 |
| Category Avg. | 46.25 | 48.42 | 60.90 | 59.72 | 57.98 | 71.01 | 54.20 | 81.97 | 80.15 | 62.64 | 69.83 | 77.50 | **87.07** | 54.75 |
| **Financial Certificate Tasks** | | | | | | | | | | | | | | |
| CFA Level 1 | 13.33 | **88.89** | 55.56 | 42.22 | 54.44 | 63.33 | 51.11 | 86.67 | 87.78 | 87.78 | 87.78 | 87.77 | 52.22 | 86.67 |
| | 0.133 | **0.889** | 0.556 | 0.418 | 0.556 | 0.631 | 0.508 | 0.867 | 0.878 | 0.878 | 0.878 | 0.878 | 0.530 | 0.867 |
| CFA Level 2 | 19.48 | **94.80** | 56.67 | 29.87 | 46.75 | 55.84 | 37.66 | 88.31 | 83.12 | 90.91 | 92.21 | **94.80** | 51.11 | 88.31 |
| | 0.199 | **0.948** | 0.567 | 0.303 | 0.485 | 0.563 | 0.383 | 0.883 | 0.835 | 0.909 | 0.922 | **0.948** | 0.519 | 0.883 |
| CFA Level 3 | 16.67 | 78.21 | 52.56 | 24.36 | 47.44 | 51.28 | 51.28 | 70.51 | 66.67 | 69.23 | **79.49** | 78.20 | 51.28 | 70.51 |
| | 0.179 | 0.782 | 0.538 | 0.271 | 0.496 | 0.517 | 0.526 | 0.705 | 0.675 | 0.697 | **0.795** | 0.782 | 0.557 | 0.705 |
| CPA REG | 31.68 | 87.13 | 63.37 | 41.58 | 65.35 | 67.33 | 45.54 | 80.20 | 88.12 | 90.10 | 90.10 | **91.08** | 51.28 | 80.20 |
| | 0.317 | 0.871 | 0.638 | 0.426 | 0.654 | 0.667 | 0.459 | 0.802 | 0.885 | 0.901 | 0.901 | **0.911** | 0.557 | 0.802 |
| Category Avg. | 20.29 | 87.26 | 57.04 | 34.51 | 53.49 | 59.44 | 46.40 | 81.42 | 81.42 | 84.50 | 87.40 | **87.96** | 51.47 | 81.42 |
| **Financial Reporting Tasks** | | | | | | | | | | | | | | |
| FiNER | 21.28 | 22.62 | 63.91 | 61.82 | 68.92 | 72.29 | 30.76 | 74.10 | 74.32 | 70.92 | 70.72 | 69.56 | **80.32** | 67.97 |
| | 0.232 | 0.266 | 0.638 | 0.606 | 0.699 | 0.725 | 0.369 | 0.759 | 0.760 | 0.732 | 0.724 | 0.757 | **0.802** | 0.694 |
| FNXL | 3.64 | 2.55 | 37.75 | 20.14 | 27.33 | 42.41 | 13.16 | 23.57 | 23.05 | 33.50 | 35.68 | 33.67 | **47.98** | 28.23 |
| | 0.045 | 0.026 | 0.356 | 0.210 | 0.288 | 0.398 | 0.116 | 0.250 | 0.253 | 0.311 | 0.348 | 0.314 | **0.438** | 0.282 |
| XBRL Term | 0.574 | 0.563 | 0.572 | 0.587 | 0.573 | 0.584 | 0.595 | 0.599 | 0.606 | 0.606 | 0.630 | 0.672 | 0.666 | **0.676** |
| Category Avg. | 27.44 | 27.16 | 52.95 | 46.89 | 51.18 | 57.70 | 34.47 | 52.52 | 52.66 | 55.01 | 56.47 | 56.81 | **64.97** | 54.60 |
| **Financial Statement Analysis Tasks** | | | | | | | | | | | | | | |
| Tag Extr. | 69.16 | 74.15 | 80.27 | 69.64 | 85.03 | 81.60 | 70.22 | **89.13** | 86.89 | 80.44 | 85.26 | 84.51 | 85.03 | 88.78 |
| | 0.739 | 0.755 | 0.811 | 0.782 | 0.849 | 0.864 | 0.739 | 0.886 | 0.872 | 0.896 | 0.845 | 0.907 | | **0.908** |
| Value Extr. | 52.46 | 74.21 | 98.02 | 88.19 | 98.01 | 97.01 | 72.29 | 98.49 | 97.14 | 98.57 | 99.13 | 98.80 | **99.20** | 97.62 |
| | 0.565 | 0.743 | 0.980 | 0.904 | 0.982 | 0.974 | 0.786 | 0.986 | 0.974 | 0.988 | **0.992** | 0.989 | **0.992** | 0.978 |
| Formula Constr. | 12.92 | 11.91 | 61.90 | 59.28 | 22.75 | 79.76 | 17.73 | 77.61 | 89.34 | 88.02 | **89.46** | 62.39 | 67.85 | 83.33 |
| | 0.201 | 0.187 | 0.644 | 0.665 | 0.315 | 0.820 | 0.233 | 0.876 | **0.898** | 0.882 | 0.893 | 0.627 | 0.786 | 0.840 |
| Formula Calc. | 27.27 | 47.62 | 53.57 | 77.49 | 85.99 | 83.59 | 33.65 | 98.68 | 92.81 | 98.92 | 98.80 | 48.50 | 54.76 | **99.04** |
| | 0.317 | 0.488 | 0.536 | 0.783 | 0.868 | 0.857 | 0.369 | 0.990 | 0.947 | **0.993** | 0.988 | 0.483 | 0.548 | **0.993** |
| Finance Bench | 0.443 | 0.584 | 0.552 | 0.528 | 0.573 | 0.564 | 0.580 | 0.511 | 0.542 | 0.477 | 0.575 | 0.617 | 0.544 | **0.621** |
| Financial Math | 11.00 | 36.00 | 19.00 | 10.50 | 21.50 | 27.00 | 32.00 | 30.00 | 26.50 | 28.50 | 34.50 | 46.00 | **66.00** | 58.00 |
| | 0.136 | 0.378 | 0.204 | 0.134 | 0.255 | 0.296 | 0.367 | 0.332 | 0.307 | 0.317 | 0.370 | 0.464 | **0.785** | 0.588 |
| Category Avg. | 36.19 | 50.38 | 61.33 | 59.65 | 61.76 | 70.89 | 47.31 | 74.17 | 74.48 | 73.69 | 77.44 | 66.98 | 71.21 | **81.48** |
| **Overall Average** (Using accuracy for classification and BERTScore F1 × 100 for generation) | | | | | | | | | | | | | | |
| Avg. | 34.64 | 53.86 | 58.97 | 52.36 | 57.16 | 66.44 | 47.27 | **74.74** | 74.29 | 69.53 | 73.82 | 73.12 | 71.08 | 68.78 |

Table 5: Comparison of fine-tuning cost. GPT-4o cost is estimated based on 4 epochs of fine-tuning at OpenAI fine-tuning pricing (OpenAI, 2025).

| Models | Time | GPUs | Est. Cost (USD) |
|---|---|---|---|
| BloombergGPT | 53 days | 512×A100 | $2.7 M |
| LoRA | 14.9h | 4 × A5000 | $15.50 |
| QLoRA | 14.1h | 4 × A5000 | $14.66 |
| DoRA | 15.9h | 4 × A5000 | $16.54 |
| rsLoRA | 14.5h | 4 × A5000 | $15.11 |
| Gemini 2.0 FL | 8.8h | - | $162.02 |
| GPT-4o-mini | - | - | $312.00 |

Figure 3: Average inference time of LoRA fine-tuned Llama 3.1 8B and LoRA fine-tuned Gemini 2.0 FL across tasks

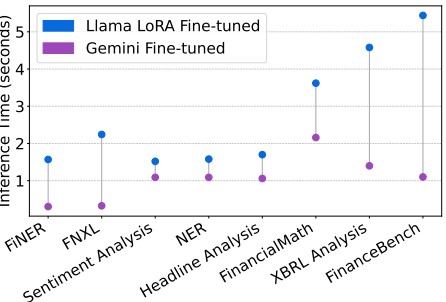

Table 6: Accuracy on MMLU, GSM8K, TriviaQA, and CoQA benchmarks for Llama 3.1 8B base and various LoRA adapters. Scores are colored relative to base: gray (same), green (higher), red (lower)

| Dataset | Llama 3.1 8B (base) | Llama 3.1 8B (FiNER) | | | | |
|---|---|---|---|---|---|---|
| | | LoRA 8bit-r8 | QLoRA 4bit-r4 | QLoRA 4bit-r32 | DoRA 8bit-r8 | rsLoRA 8bit-r8 |
| **MMLU** | 0.229 | 0.229 | 0.229 | 0.229 | 0.229 | 0.229 |
| **GSM8K** | 0.011 | 0.011 | 0.014 | 0.010 | 0.011 | 0.016 |
| **TriviaQA-Open (F1)** | 0.667 | 0.658 | 0.673 | 0.666 | 0.663 | 0.663 |
| **CoQA (F1)** | 0.711 | 0.709 | 0.697 | 0.715 | 0.707 | 0.715 |

### 4.3 Angle III: Resource Usage and Performance Trade-offs of LoRA methods

Table 5 details the computational costs of LoRA fine-tuned models. Using four NVIDIA A5000 GPUs, the wall-clock time for fine-tuning ranged from 14.1 hours (QLoRA) to 15.9 hours (DoRA), corresponding to a total of approximately 56.4 to 63.6 GPU hours. At an estimated rate of $0.26 per GPU hour, this translates to a cost of roughly $14.66 to $16.54. This is substantially more cost-effective than fine-tuning services from providers like Google or OpenAI. Figure 3 illustrates the inference time of fine-tuned models on various datasets. Gemini API generally exhibits lower inference latency and is less sensitive to increasing prompt lengths than local Llama 3.1 8B Instruct inference, even when accounting for network overhead for the API. However, the inference speed of locally deployed Llama models can be significantly enhanced through the use of larger batch sizes.

### 4.4 Angle IV: Practicability of Applying LoRA in Real-world Financial Scenarios

**Federated LoRA** The sensitive nature of financial data necessitates privacy-preserving techniques like Federated Learning for collaborative training. To explore this, we evaluated Federated LoRA (Sun et al., 2024), with results presented in Table 7. Our experimental setup simulated a four-node environment employing the FedAvg algorithm (McMahan et al., 2017), where the sentiment analysis dataset was partitioned across these nodes. The performance of this approach was benchmarked against both the base Llama model and standard centralized LoRA fine-tuning. While Federated LoRA did not match the performance levels of centralized LoRA, the results demonstrate a notable improvement compared to the base Llama model.

**Catastrophic Forgetting** A major concern with PEFT is that fine-tuning on domain-specific tasks leads to the model forgetting pre-training knowledge. To investigate this, we evaluated eight adapters—covering both sentiment and FiNER tasks and all four LoRA variants—as well as the Llama 3.1 8B Instruct base model on two out-of-domain benchmarks, MMLU (Hendrycks et al., 2020) and GSM8K (Cobbe et al., 2021). We used a zero-shot, no chain-of-thought setting to isolate stored knowledge. Table 6 shows identical MMLU accuracy across all adapters and the base model,

Table 7: Performance comparison of central LoRA and LoRA federated learning (8 bit, rank 8) on Llama 3.1 8B using four nodes: accuracy (blue) and F1 score (gray).

| | Sentiment | | | | XBRL Analysis | | | |
|---|---|---|---|---|---|---|---|---|
| | FPB | FiQA SA | TFNS | NWGI | Tag Extr. | Value Extr. | Formula Constr. | Formula Calc. |
| Base | 68.73 | 46.55 | 24.16 | 43.86 | 69.16 | 52.46 | 12.92 | 27.27 |
| | 0.677 | 0.557 | 0.339 | 0.538 | 0.739 | 0.565 | 0.201 | 0.317 |
| Central (LoRA) | **85.64** | **81.28** | **88.02** | 54.16 | **89.13** | **98.49** | **77.61** | **98.68** |
| | **0.922** | **0.884** | **0.932** | 0.690 | **0.886** | **0.986** | **0.876** | **0.990** |
| FedAvg (McMahan et al., 2017) | 82.43 | 76.17 | 73.41 | **56.02** | 69.05 | 79.76 | 13.10 | 28.57 |
| | 0.902 | 0.860 | 0.842 | **0.698** | 0.745 | 0.814 | 0.170 | 0.305 |

and equal or higher scores on GSM8K. Hence, at the ranks $r$ we tested (4 and 8) with $\alpha{:}r$ equal to 8:1 or 4:1, we observe that LoRA does not exhibit catastrophic forgetting. In fact, the slight GSM8K performance improvements hint at cross-domain knowledge transfer—fine-tuning on financial data may improve the model's numerical reasoning skills.

## 5 RELATED WORKS

### 5.1 FINANCIAL LLMS AND BENCHMARKS

BloombergGPT (Wu et al., 2023), a 50B parameter model trained from scratch on mixed financial and general data, was the first specialized financial LLM, showing strong performance on domain-specific tasks. In contrast, FinGPT (Liu et al., 2023a; 2024f;e) applies LoRA fine-tuning to open-source LLMs, an efficient alternative that improves performance over base models, even surpassing BloombergGPT. Financial benchmarks like FinBen (Xie et al., 2024) and PIXIU (Xie et al., 2023) show that while current LLMs handle textual analysis well, they struggle with complex financial reasoning.

### 5.2 PARAMETER-EFFICIENT FINE-TUNING (PEFT)

To mitigate the high cost of full fine-tuning, Parameter-efficient fine-tuning (PEFT) methods update only a small subset of model parameters (Mao et al., 2024). A prominent PEFT technique is Low-rank Adaptation (LoRA) (Hu et al., 2022), which injects trainable low-rank matrices and achieves performance comparable to full fine-tuning. Quantized LoRA (QLoRA) (Dettmers et al., 2023) further enhances efficiency by applying LoRA to a 4-bit quantized model, drastically reducing memory usage.

### 5.3 LORA WITH FEDERATED LEARNING

Federated learning is necessary to fine-tune LLMs on decentralized private data, which is common in finance. Consequently, methods combining LoRA with federated learning have been proposed, such as PrivateLoRA (Wang et al., 2023) and FFA-LoRA (Sun et al., 2024).

## 6 CONCLUSION AND FUTURE WORK

In this paper, we present FinLoRA, a benchmark that evaluates LoRA methods on both general and highly specialized financial tasks. We curated 19 diverse datasets covering a wide range of financial applications. Our study includes 59 rounds of fine-tuning and ~300 rounds of evaluation to thoroughly assess and analyze commonly used LoRA methods. FinLoRA offers insights into overall performance, task-specific results, resource requirements for fine-tuning and inference, and practical considerations for real-world deployment—including data privacy in collaborative training and catastrophic forgetting. Our results demonstrate that fine-tuning can significantly enhance the effectiveness of LLMs on financial tasks. Additionally, FinLoRA provides a comprehensive collection of datasets with baseline results, laying a solid foundation for future research in this field. Moving forward, we plan to expand FinLoRA by incorporating additional LoRA methods into the project.

REPRODUCIBILITY STATEMENT

To ensure the reproducibility of our results, all code, datasets, and pretrained LoRA adapters have been made publicly and anonymously available at https://anonymous.4open.science/r/FinLoRA-ICLR-115E/. The repository includes detailed documentation and README files with instructions for setting up the environment and running the experiments. Appendix C includes details on experimental setup, such as hyperparameters. Our XBRL analysis dataset and project code are also available in supplementary materials on Open Review. Furthermore, a comprehensive description of our novel XBRL analysis dataset, including all data collection and processing steps, is provided in Appendix E.

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

# Appendix

## A    DATASETS

### A.1    OVERVIEW

We aim to assess models' capabilities in four key areas of financial expertise—general financial tasks, financial certificate tasks, financial reporting tasks, and financial statement analysis tasks.

### A.2    TASK DETAILS

In this subsection, we describe the different tasks on which we benchmarked models.

#### A.2.1    GENERAL FINANCIAL TASKS

General financial tasks include tasks commonly used to benchmark financial LLMs, such as BloombergGPT Wu et al. (2023) and FinGPT Liu et al. (2023b). We aim to enhance and benchmark our model with three main general capabilities: sentiment analysis, headline analysis, and name entity recognition.

**Sentiment Analysis**    The following datasets all consist of input texts from various sources like news or tweets and sentiment labels such as "positive", "negative", and "neutral".

- **Financial Phrase Bank (FPB)** Malo et al. (2013) contains sentiment-annotated sentences extracted from financial news and reports. We manually created the train/test split with 25% train.
- **Financial question-answering sentiment analysis (FiQA SA)** Maia et al. (2018) is another sentiment analysis dataset with the same labels as FPB from microblog headlines and financial news.
- **Twitter financial news sentiment (TFNS)** Rahman (2022) comprises annotated tweets related to financial news labeled with sentiment categories.
- **News with GPT instruction (NWGI)** Liu et al. (2024e) comprises samples with seven labels ranging from "strong negative" to "strong positive".

**Headline classification**    The Headline dataset Sinha & Khandait (2020) classifies headlines based on various questions into two classes: "yes" and "no".

**Named entity recognition (NER)**    The NER dataset Salinas Alvarado et al. (2015) annotates one entity per sentence, categorized into one of three classes: "location", "person", and "organization"

#### A.2.2    FINANCIAL CERTIFICATE TASKS

To measure LLMs' performance in financial analysis and regulation compliance, we introduced questions from three financial analyst certification exams and one accountant certification exam. The questions we test models on are from the three levels of Chartered Financial Analyst (CFA) exams as well as the Certified Public Accountant (CPA) REG (Regulation) exam. We manually created the train/test split. We combined the train sets from each exam into one question set to better fine-tune Llama 3.1 8B Instruct due to the overlapping topics of questions. The test sets remain as four separate question sets, with CFA Level I to III and CPA Regulation.

#### A.2.3    FINANCIAL REPORTING TASK

Financial reporting refers to the tasks of creating financial reports that are in compliance with U.S. Securities and Exchange Commission (SEC) regulations or other regulatory agencies around the world. Specifically, the SEC requires publicly traded companies to file financial reports in XBRL formats using the U.S. Generally Accepted Accounting Principles (GAAP), an accounting standard.

**Terminology**   XBRL Term Han et al. (2024) includes over 6,000 XBRL terminology entries and their explanations from the XBRL International website. The train/test split was sampled randomly.

**Tagging**   XBRL tagging is a key step in creating XBRL reports, where numerical entities in a financial statement such as earning calls must be tagged with US GAAP tags. We use two existing datasets to benchmark tagging performance. FiNER Loukas et al. (2022) includes train and test sets of sentences annotated with 139 types of XBRL tags. We processed the dataset using the batched approach so each input comprises tag options and four questions of highlighted entity tagging, and the answer includes the four correct tags. FNXL Sharma et al. (2023) (Financial Numeric Extreme Labeling) is similar to FiNER but with 2,794 labels. Only a test split was published by the authors and used for our benchmark, so it was tested on the models that were fine-tuned on FiNER. We processed the dataset with a QA style prompt and batched approach to improve efficiency and reduce token usage. The input includes multiple (e.g., four) numerical entities to be tagged simultaneously, and a list of potential US GAAP tag options is provided. Providing the options enables the model to understand which tags are valid choices without needing pre-existing knowledge of the entire taxonomy. The model can infer the appropriate tag from its name and context.

### A.2.4   FINANCIAL STATEMENT ANALYSIS TASKS

Financial statement analysis refers to the task of extracting and analyzing information from financial statements or reports. We include three main sub-tasks: financial math, XBRL analysis, and FinanceBench.

**Financial Math**   Financial math Han et al. (2024) is the basics of financial analysis. It tests the model's capabilities in applying financial formulas to questions. The dataset includes 50 formulas, each with 20 questions. We created a manual train/test split with 16 of those 20 questions for each formula as the train split and the rest as the test split.

**XBRL Analysis**   One approach for financial statement analysis is to directly examine XBRL reports in their raw form. We created the XBRL extraction dataset. The QA style dataset consists of questions and answers derived from XBRL filings from 2019 to 2023 for Dow Jones 30 companies. Each example includes a question, a text segment from an XBRL file containing the answer, and the ground truth. The prompt consists of a portion of a filtered XBRL report in XML and the question.

- **XBRL tag extraction** involves extracting a specific XBRL tag from an XBRL raw text segment given a natural language description of the tag.
- **XBRL value extraction** focuses on extracting a numeric value from the raw XBRL text segment given a natural language description of the value.
- **XBRL formula construction** retrieves multiple tags and outputs a financial formula.
- **XBRL formula calculation** is similar to formula construction but with the goal of outputting formulas substituted with numerical values.

To allow better instruction following for the base model, we use one-shot prompting by providing an example question and answer.

**FinanceBench**   FinanceBench Islam et al. (2023) is another approach for financial analysis which reads the financial reports in PDF format. We include 139 of the 150 questions (some financial report PDFs are no longer accessible) and created a manual train/test split. During dataset pre-processing, we use Google Cloud Document OCR to convert relevant PDF pages to text.

### A.3   TASK SIMILARITY

To further measure the correlation between in-category task similarity and multi-task performance, we estimate the in-category task similarity by averaging the cosine similarity between each pair of LoRA adapters within each task category. The cosine similarity is computed by first calculating the effective weight update matrix ($\Delta W = BA$) for each attention layer in a LoRA adapter. These matrices are then flattened into vectors. For any two tasks, we compute the cosine similarity between their

corresponding vectors for each shared attention layer and then average these layer-wise similarities. The final value in the table is the average of these pairwise similarities across all tasks within the category.

The final task similarity and the average single-task and multi-task performance (and their difference) are displayed below.

Table 8: Correlation between Task Similarity and Multi-Task Performance

| Task Category | Avg. Cos. Similarity | Avg. Single-Task Perf. | Avg. Multi-Task Perf. | Avg. Perf. $\Delta$ (Multi - Single) |
|---|---|---|---|---|
| General Financial | 0.0043 | 81.97 | 54.75 | -27.22 |
| Financial Reporting | 0.0048 | 52.52 | 54.60 | +2.08 |
| Financial Statement Analysis | **0.0108** | 74.17 | 81.48 | **+7.31** |

The results show a clear correlation between the similarity of LoRA adapters and the success of multi-task learning.

- **Positive Transfer:** The Financial Statement Analysis tasks show the highest average similarity (0.0108). This corresponds to successful multi-task learning, where the combined model outperformed the average of single-task models by 7.31 points. This suggests the tasks in this category are complementary, and their learned parameter updates are constructive.

- **Negative Interference:** The General Financial and Financial Reporting tasks have low similarity scores (0.0043 and 0.0048). This dissimilarity leads to interference in the multi-task setting, causing a significant performance drop of 27.22 points for General Financial Tasks and only 2.08 points increase in Financial Reporting Tasks compared to their single-task counterparts. This indicates that the parameter updates required for these tasks are conflicting or marginally constructive.

### A.4 INPUT/OUTPUT EXAMPLES

In this subsection, we provide input and output examples for the tasks on which we benchmarked models. We only provide the prompt templates without question or answer choice text for financial certificate tasks due to copyright concerns. Sample CFA and CPA REG questions can be found online.

### A.4.1 GENERAL FINANCIAL TASKS

| Dataset | **Financial Phrase Bank (FPB) Malo et al. (2013)** |
|---|---|
| Task | Sentiment Analysis |
| Input | `Instruction: What is the sentiment of this news?` `Please choose an answer from {negative/neutral/positive}.` `Input: Pharmaceuticals group Orion Corp reported a fall` `in its third-quarter earnings that were hit by larger` `expenditures on R&D and marketing. Answer:` |
| Output | `negative` |

| Dataset | **FiQA SA Maia et al. (2018)** |
|---|---|
| Task | Sentiment Analysis |
| Input | `Instruction: What is the sentiment of this news?` `Please choose an answer from {negative/neutral/positive}.` `Input: Johnson Matthey raises prospect of investor` `payout Answer:` |
| Output | `positive` |

| Dataset | **Twitter Financial News Sentiment (TFNS) Rahman (2022)** |
|---|---|
| Task | Sentiment Analysis |
| Input | What is the sentiment of this tweet?  Please choose an answer from {negative/neutral/positive}.Input: $BYND - JPMorgan reels in expectations on Beyond Meat https://t.co/bd0xbFGjkT Answer: |
| Output | negative |

| Dataset | **News with GPT (NWGI)** |
|---|---|
| Task | Sentiment Analysis |
| Input | Instruction:  What is the sentiment of this news? Please choose an answer from {strong negative/moderately negative/mildly negative/neutral/mildly positive/moderately positive/strong positive}.  Input: Amid a soft performance for the major equity indices on Tuesday, Nvidia (NASDAQ: NVDA) posted a particularly glaring loss.  Shares continued to fall in sympathy with fellow semiconductor specialist Micron Technology (NASDAQ: MU) following its disappointing earnings results last week.  Answer: |
| Output | moderately negative |

| Dataset | **Financial Headline Analysis Sinha & Khandait (2020)** |
|---|---|
| Task | Headline Analysis |
| Input | Instruction:  Does the news headline talk about price? Please choose an answer from Yes/No.  Input:  Gold futures edge up after two-session decline Answer: |
| Output | No |

| Dataset | **Named Entity Recognition (NER) Salinas Alvarado et al. (2015)** |
|---|---|
| Task | Named Entity Recognition |
| Input | Instruction:  What is the entity type of '40 William St' in the input sentence.  Options:  person, location, organization Input:  This LOAN AND SECURITY AGREEMENT dated January 27, 1999, between SILICON VALLEY BANK (" Bank "), a California - chartered bank with its principal place of business at 3003 Tasman Drive, Santa Clara, California 95054 with a loan production office located at 40 William St., Ste.  Answer: |
| Output | location |

A.4.2 FINANCIAL CERTIFICATE TASKS

| Dataset | **CFA Level I** |
|---------|-----------------|
| Task | Question Answering |
| Input | Please answer the following question with the exact letter and choice text and no explanations that you choose. Nothing else like an explanation should be in your response.
Question: [CFA Level I context and question omitted due to copyright]
Choices: [Question choices omitted due to copyright] |
| Output | [Letter of selected answer with answer text omitted due to copyright] |

| Dataset | **CFA Level II** |
|---------|------------------|
| Task | Question Answering |
| Input | Look at the following context.
[CFA Level II question-set-level context omitted due to copyright]
Based on this, please answer the following question with the exact letter and choice text and no explanations that you choose. Nothing else like an explanation should be in your response.
Question: [CFA Level II question omitted due to copyright]
Choices: [Question choices omitted due to copyright] |
| Output | [Letter of selected answer with answer text omitted due to copyright] |

| Dataset | **CFA Level III** |
|---------|-------------------|
| Task | Question Answering |
| Input | Look at the following context.
[CFA Level III question-set-level context omitted due to copyright]
Based on this, please answer the following question with the exact letter and choice text and no explanations that you choose. Nothing else like an explanation should be in your response.
Question: [CFA Level III question omitted due to copyright]
Choices: [Question choices omitted due to copyright] |
| Output | [Letter of selected answer with answer text omitted due to copyright] |

| Dataset | **CPA REG** |
|---------|-------------|
| Task | Question Answering |
| Input | Please answer the following question with the exact letter and choice text and no explanations that you choose. Nothing else like an explanation should be in your response.
Question: [CPA REG question omitted due to copyright]
Choices: [Question choices omitted due to copyright] |
| Output | [Letter of selected answer with answer text omitted due to copyright] |

### A.4.3 FINANCIAL REPORTING TASKS

| Dataset | **XBRL Term** Han et al. (2024) |
|---|---|
| Task | XBRL Terminology |
| Input | Explain this XBRL term briefly in one sentence: Frequent Flier Liability, Current.  Answer: |
| Output | Value of revenue deferred or cost to provide future products or services, primarily air transportation, associated with programs used by airlines to encourage passenger loyalty by providing rewards geared to the frequency of travel on the sponsoring airline, typically in the form of frequent flyer miles, points, or segments that can be accumulated and converted into free or discounted travel or other redemption options.  Used to reflect the current portion of the liability (within one year or within the normal operating cycle if longer) |

| Dataset | **FiNER Loukas et al. (2022)** |
|---|---|
| Task | XBRL Tagging |
| Input | You are XBRL expert.  Here is a list of US GAAP tags options:  [omitted for this table] Answer the following 4 independent questions by providing only 4 US GAAP tags answers in the order of the questions.  Each answer must be separated by a comma (,).  Provide nothing else. 1.  What is best tag for entity "66,32" in sentence: "Unvested restricted stock outstanding as of March 31, 2020 and 2019 were 42,690 and 66,321 shares, respectively.?" 2.  What is best tag for entity "625,000" in sentence: "Unvested AO LTIP Units outstanding as of March 31, 2020 and 2019 were 625,000 and 625,000, respectively.?" 3.  What is best tag for entity "625,000" in sentence: "Unvested AO LTIP Units outstanding as of March 31, 2020 and 2019 were 625,000 and 625,000, respectively.?" 4.  What is best tag for entity "0.20" in sentence: "Distributions declared per common unit for each of the three month periods ended March 31, 2020 and 2019 was $ 0.20 per unit.?" Output US GAAP tags:"" |
| Output | ShareBasedCompensationArrangementByShareBasedPaymentAward EquityInstrumentsOtherThanOptionsNonvestedNumber, ShareBasedCompensationArrangementByShareBasedPaymentAwardEquity InstrumentsOtherThanOptionsNonvestedNumber,ShareBasedCompensation ArrangementByShareBasedPaymentAwardEquityInstrumentsOtherThanOptions NonvestedNumber,CommonStockDividendsPerShareDeclared |

| Dataset | **FiNER Loukas et al. (2022)** |
|---|---|
| Task | XBRL Tagging |
| Input | You are XBRL expert. Here is a list of US GAAP tags options: [omitted for this table] Answer the following 4 independent questions by providing only 4 US GAAP tags answers in the order of the questions. Each answer must be separated by a comma (,). Provide nothing else. 1. What is best tag for entity "66,32" in sentence: "Unvested restricted stock outstanding as of March 31, 2020 and 2019 were 42,690 and 66,321 shares, respectively.?" 2. What is best tag for entity "625,000" in sentence: "Unvested AO LTIP Units outstanding as of March 31, 2020 and 2019 were 625,000 and 625,000, respectively.?" 3. What is best tag for entity "625,000" in sentence: "Unvested AO LTIP Units outstanding as of March 31, 2020 and 2019 were 625,000 and 625,000, respectively.?" 4. What is best tag for entity "0.20" in sentence: "Distributions declared per common unit for each of the three month periods ended March 31, 2020 and 2019 was $ 0.20 per unit.?" Output US GAAP tags:"" |
| Output | ShareBasedCompensationArrangementByShareBasedPaymentAward EquityInstrumentsOtherThanOptionsNonvestedNumber, ShareBasedCompensationArrangementByShareBasedPaymentAwardEquity InstrumentsOtherThanOptionsNonvestedNumber,ShareBasedCompensation ArrangementByShareBasedPaymentAwardEquityInstrumentsOtherThanOptions NonvestedNumber,CommonStockDividendsPerShareDeclared |

| Dataset | **FNXL Sharma et al. (2023)** |
|---|---|
| Task | XBRL Tagging |
| Input | You are XBRL expert.  Choose the best XBRL US GAAP tag for each highlighted entity in the sentences below. Provide only the US GAAP tags, comma-separated, in the order of the sentences and highlighted entity.  Provide nothing else.  [Tags omitted for this table]
1.  What is the best us gaap tag for entity "6.3" in sentence:  "The projected benefit obligation and fair value of plan assets for U.S. pension plans with projected benefit obligations in excess of plan assets was $6.3 billion and $4.7 billion, respectively, as of December31, 2019 and $5.5 billion and $4.1 billion, respectively, as of December31, 2018."?
2.  What is the best us gaap tag for entity "124,043" in sentence:  "Capitalized software, net of accumulated amortization of $124,043 in 2020 and $104,237 in 2019"?
3.  What is the best us gaap tag for entity "1.5" in sentence:  "The Company purchased 30 million and 57 million shares under stock repurchase programs in fiscal 2020 and 2019 at a cost of $1.5 billion and $3.8 billion, respectively."?
4.  What is the best us gaap tag for entity "651,313" in sentence:  "This multi-tenant mortgage loan is interest-only with a principal balance due on maturity, and it is secured by seven properties in six states, totaling approximately 651,313 square feet."? |
| Output | DefinedBenefitPlanPensionPlanWithProjectedBenefitObligation InExcessOfPlanAssetsProjectedBenefitObligation, CapitalizedComputerSoftwareAccumulatedAmortization, PaymentsForRepurchaseOfCommonStock, AreaOfRealEstateProperty |

## A.4.4 FINANCIAL STATEMENT ANALYSIS

| Dataset | **XBRL Tags Extraction** |
|---|---|
| Task | XBRL Analysis |
| Input | You are a knowledgeable XBRL assistant.  Your task is to analyze the XBRL context and provide an accurate and very concise answer to the question, The example question can help you to learn the answer format. DO NOT output xml, code, explanation or create new question.
Example question:  What is the US GAAP XBRL tag for Cash and Cash Equivalents as reported by Example Company Inc for the Fiscal Year ending in FY 2022
Example answer:  us-gaap:AnExampleTagName

XML File:  [XBRL omitted]

Question:  What is the US GAAP XBRL tag for Total Equity as reported by Home Depot Inc for the Fiscal Year ending in FY 2023?
Answer: |
| Output | us-gaap:StockholdersEquity |

| Dataset | **XBRL Values Extraction** |
|---------|---------------------------|
| Task | XBRL Analysis |
| Input | You are a knowledgeable XBRL assistant. Your task is to analyze the XBRL context and provide an accurate and very concise answer to the question, The example question can help you to learn the answer format. DO NOT output xml, code, explanation or create new question. Example question: What is the value of Example company's income for the Fiscal year ending in FY 2020? Example answer: 80000000 XML File: [XBRL omitted] Question: What is the value of Unitedhealth Group Inc's Total Revenue for the Fiscal Year ending in FY 2023? Answer: |
| Output | 371622000000.0 |

| Dataset | **XBRL Formula Construction** |
|---------|-------------------------------|
| Task | XBRL Analysis |
| Input | You are a knowledgeable XBRL assistant. Your task is to analyze the XBRL context and provide an accurate and very concise answer to the question, The example question can help you to learn the answer format. DO NOT output xml, code, explanation or create new question. Example question: What is the formula for the Gross Profit Margin of Example Inc, formatted with the relevant US GAAP XBRL tags Example answer: us-gaap:ExampleTag / us-gaap:AnotherExampleTag) * 100 XML File: [XBRL omitted] Question: Can you provide the formula of the Cash Flow Margin for Microsoft Corp, using the appropriate US GAAP XBRL tags, as it applies to the FY 2023? * 100') Answer: |
| Output | (us-gaap:NetCashProvidedByUsedInOperatingActivities / us-gaap:RevenueFromContractWithCustomerExcludingAssessedTax) * 100 |

| Dataset | **XBRL Formula Calculation** |
|---|---|
| Task | XBRL Analysis |
| Input | You are a knowledgeable XBRL assistant. Your task is to analyze the XBRL context and provide an accurate and very concise answer to the question, The example question can help you to learn the answer format. DO NOT output xml, code, explanation or create new question. Example question: Can you provide the formula for Operating Profit Margin from Example Corp for the Fiscal Year ending in FY 2022? Example answer: (50000000 / 3590000000) * 100 XML File: [XBRL omitted] Question: Can you provide the value for Net Profit Margin from Apple Inc for the Fiscal Year ending in FY 2023? Answer with a formula substituted with values. Answer: |
| Output | (96995000000 / 383285000000) * 100 |

| Dataset | **Financial Math Han et al. (2024)** |
|---|---|
| Task | Math |
| Input | Use formula Operating Cash Flow (OCF) to answer the question. Answer with a numerical answer with 2 decimal places and with no explanations or other text. Formula: OCF = Net Income + Depreciation + Changes in Working Capital, Explanation: Net Income: Company's total earnings, Depreciation: Depreciation expense, Changes in Working Capital: Change in current assets minus current liabilities. Question: "What is the Operating Cash Flow when a company's financial sheets reflect a Net Income of $260,000, Depreciation of $48,000, and a $35,000 decline in Working Capital?". Answer: |
| Output | 343000.0 |

| Dataset | **FinanceBench Han et al. (2024)** |
|---|---|
| Task | XBRL QA |
| Input | Answer the following question very briefly using scanned text from pages. No explanation needed. Question: Has AMCOR's quick ratio improved or declined between FY2023 and FY2022? If the quick ratio is not something that a financial analyst would ask about a company like this, then state that and explain why. Document Pages Context: [XBRL omitted] Answer: |
| Output | The quick ratio has slightly improved from 0.67 times to 0.69 times between FY 2023 and FY 2022.(3.4% jump) |

# B  LoRA Foundations and Methods

## B.1  What is LoRA?

LoRA is a technique to efficiently update the parameters of pre-trained language models when fine-tuning on new tasks.

## B.2  Foundations of LoRA

In this subsection, we introduce two fundamental concepts needed to understand LoRA—ranks and fine-tuning.

### B.2.1  Ranks

Rank is the number of linearly independent rows or columns in a matrix. Linearly independent columns, for example, are columns whose entries cannot be written as an integer-weighted sum of earlier columns.

$$W = \begin{bmatrix} 1 & 7 & 2 & 8 & 5 \\ 2 & 10 & 4 & 12 & 10 \\ 3 & 15 & 12 & 18 & 27 \\ 4 & 12 & 16 & 16 & 36 \end{bmatrix}, \qquad \text{Dimensions: } 4 \times 5 \text{ (rows} \times \text{columns)}$$

In this matrix there are **two** linearly independent columns, so $\text{rank}(W) = 2$.

- Column 1 is independent (nothing precedes it).
- Column 2 cannot be written as a multiple of Column 1, so it is also independent.
- Columns 3–5 are dependent:

$$C_3 = 2C_1, \qquad C_4 = C_1 + C_2, \qquad C_5 = C_1 + 2C_2.$$

Re-expressing those dependencies in vector form:

$$W = \underbrace{\begin{bmatrix} 1 & 7 \\ 2 & 10 \\ 3 & 15 \\ 4 & 12 \end{bmatrix}}_{B \in \mathbb{R}^{4 \times 2}} \underbrace{\begin{bmatrix} 1 & 0 & 2 & 1 & 1 \\ 0 & 1 & 0 & 1 & 2 \end{bmatrix}}_{A \in \mathbb{R}^{2 \times 5}}.$$

$$\text{Dimensions}(W) = d \times k = 4 \times 5,$$
$$\text{Dimensions}(B) = d \times r = 4 \times 2,$$
$$\text{Dimensions}(A) = r \times k = 2 \times 5,$$
$$\text{Dimensions}(BA) = d \times k = \text{Dimensions}(W).$$

$$\text{Parameters}(W) = 4 \times 5 = 20,$$
$$\text{Parameters}(B) = 4 \times 2 = 8,$$
$$\text{Parameters}(A) = 2 \times 5 = 10,$$
$$\text{Parameters}(BA) = 8 + 10 = 18.$$

Thus storing $B$ and $A$ uses fewer parameters than storing $W$ directly—a key idea behind *low-rank adaptation* (LoRA).

### B.2.2  Full Fine-Tuning

Consider a pre-trained model M with 500 million parameters. Suppose we pre-trained M with two tasks. Task 1 is Masked Language Modeling (MLM), where we mask some words in a sentence, and the task is to predict the sentence with the masked tokens filled in. Task 2 is Next Sentence Prediction

(NSP), where the task is to predict if, given 2 sentences, whether or not sentence A comes before sentence B.

If we want to fine-tune the pre-trained model M on a new task, Named Entity Recognition (NER), where the task is to annotate one entity (location/person/organization) per sentence in a financial task.

When we perform full fine-tuning on model M, all parameters are updated based on the gradients we compute during backpropagation. In backpropagation, we compute the loss (the difference between the predicted output and the target output) and propagate the loss backward through the model. As we propagate the loss backward, we compute the gradient of the loss with respect to each parameter. The optimizer uses these gradients to update the model's parameters.

If we want to fine-tune model M on another task Financial Phrase Bank (FPB), where the task is to annotate sentences from financial news and reports with sentiment, we still need to update all 500 million parameters. This is costly and can lead to overfitting and the model forgetting pre-training tasks.

### B.2.3 FINE-TUNING WITH ADAPTERS (PARAMETER-EFFICIENT FINE-TUNING—PEFT)

Parameter Efficient Fine-Tuning (PEFT) adds small adapter layers per transformer block as shown below. Let's consider a scenario in which we use PEFT to fine-tune the pre-trained model M and add two adapter layers per transformer layer.

Now, when we fine-tune M on NER, only the adapter parameters are updated. This only consists of a tiny fraction of the original parameters. The rest of the model's parameters are frozen. This means that, during backpropagation, the gradients of loss pass through them, but the parameters aren't updated. While we do have to swap the adapters and store the updated parameters separately for FPB, the number of parameters required to fine-tune on FPB is much smaller than full fine-tuning.

### B.3 LORA METHODS

In this subsection, we introduce the five LoRA methods we use in our paper. We choose LoRA Hu et al. (2022) and QLoRA Dettmers et al. (2023) due to their standard use in fine-tuning. We choose DoRA Liu et al. (2024d) and rsLoRA Kalajdzievski (2023) due to their performance enhancements: DoRA proposes fine-grained updates for achieving accuracy through LoRA, and rsLoRA proposes a scaling factor to achieve gradient stability. Lastly, we choose LoRA with federated learning for its practical ability to allow financial institutions to collaborate in fine-tuning models while using private, confidential data.

### B.3.1 LOW-RANK ADAPTATION (LORA)

LoRA adds a scaled low-rank update $\Delta \boldsymbol{W} = \gamma_r \boldsymbol{B}\boldsymbol{A}$—where $\gamma_r$ is a scaling factor ($\gamma_r = \frac{\alpha}{r}$ with $\alpha > 0$ and rank $r > 0$), $\boldsymbol{B} \in \mathbb{R}^{d \times r}$, and $\boldsymbol{A} \in \mathbb{R}^{r \times k}$—to the frozen pre-trained weight matrix $\boldsymbol{W}_0 \in \mathbb{R}^{d \times k}$.

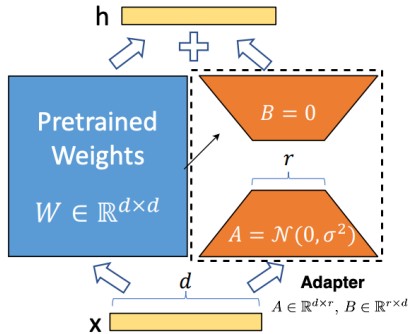

Figure 4: LoRA mechanism Hu et al. (2022).

For each multi-head attention layer, we have query, key, and value weight matrices, which we can factorize as follows:

$$W_Q^{(n)} = B_Q^{(n)} A_Q^{(n)}, \quad W_K^{(n)} = B_K^{(n)} A_K^{(n)}, \quad W_V^{(n)} = B_V^{(n)} A_V^{(n)}.$$

During fine-tuning, the weight matrices are updated as follows with the scaled low-rank update:

$$W_{Q,\text{new}}^{(n)} = W_{Q,\text{old}}^{(n)} + \gamma_r B_Q^{(n)} A_Q^{(n)},$$
$$W_{K,\text{new}}^{(n)} = W_{K,\text{old}}^{(n)} + \gamma_r B_K^{(n)} A_K^{(n)},$$
$$W_{V,\text{new}}^{(n)} = W_{V,\text{old}}^{(n)} + \gamma_r B_V^{(n)} A_V^{(n)}.$$

Because the update is in-place, no extra layers are added, and inference latency is unchanged.

### B.3.2 QUANTIZED LoRA (QLoRA)

Table 9: Finetuning LLMs with QLoRA methods: listing the number of parameters and GPU memory.

| Models | Parameters | GPU Memory Batch = 4 | GPU Memory Batch = 8 | Model size |
|---|---|---|---|---|
| Llama-3.1-8B-16bit (base) | 8.03 B | - | - | 16.06 GB |
| Llama-3.1-8B-r8-16bit | 4.72 M | 30.91 GB | 30.91 GB | 16.08 GB |
| Llama-3.1-8B-r8-8bit | 4.72 M | 11.41 GB | 11.81 GB | 8.04 GB |
| Llama-3.1-8B-r8-4bit | 4.72 M | 8.26 GB | 8.65 GB | 4.02 GB |
| Llama-3.1-8B-r4-16bit | 2.36 M | 30.90 GB | 30.90 GB | 16.07 GB |
| Llama-3.1-8B-r4-8bit | 2.36 M | 11.40 GB | 11.78 GB | 8.03 GB |
| Llama-3.1-8B-r4-4bit | 2.36 M | 8.25 GB | 8.61 GB | 4.02 GB |
| Llama-3.1-70B-16bit (base) | 70.56 B | - | - | 151.53 GB |
| Llama-3.1-70B-r8-16bit | 22.28 M | - | - | 151.57 GB |
| Llama-3.1-70B-r8-8bit | 22.28 M | 173.57 GB | 258.17 GB | 75.79 GB |
| Llama-3.1-70B-r8-4bit | 22.28 M | 42.78 GB | 42.78 GB | 37.89 GB |
| Llama-3.1-70B-r4-16bit | 11.14 M | - | - | 151.55 GB |
| Llama-3.1-70B-r4-8bit | 11.14 M | 173.36 GB | 258.11 GB | 75.70 GB |
| Llama-3.1-70B-r4-4bit | 11.14 M | 42.73 GB | 42.73 GB | 37.89 GB |

When fine-tuning, LoRA requires a large amount of GPU memory. To solve this issue, we can use QLoRA. QLoRA drastically reduces memory usage and lets you fine-tune on a single GPU.

In QLoRA, we quantize the weights of the adapter layers, reducing both parameter count and memory usage. Quantization is a technique that reduces the precision of the weights to reduce the number of bits used to store them. It consists of two parts: Rounding to the nearest integer and truncating to remove the decimal portion of a floating point number. QLoRA specifically uses 4-bit NormalFloat (NF4), an optimal data type for normally distributed weights, quantization. Pre-trained weights are usually normally distributed and centered around 0, which is why NF4 is ideal for quantization.

If we quantize from Float16 to Int4, we can represent 16 different values (bins) because Int4 has 4 bits and $2^4 = 16$. Inputs are usually normalized from -1 to 1. Very close together values, however, will be mapped to the same bin. This means that the precision is lost if we want to convert back to Float16. However, we can use blockwise quantization, where we divide the input range into blocks and quantize each block separately. QLoRA uses a 64 blocksize for better precision.

Since regular quantization relies on the bins being equally probable, QLoRA uses NormalFloat where the bins are weighted by the normal distribution. The spacing between bins is therefore closer together near 0 and further apart further away from 0.

Each block in QLoRA has a quantization constant. QLoRA employs double quantization, where it quantizes the quantization constants themselves to further save space.

The last part of QLoRA is paged optimizers. Paged optimizers reduce GPU memory spikes by switching pages to CPU memory when GPU RAM becomes full when processing long sequences, and the pages are not needed for the current computation of the forward/backward pass.

The forward pass for QLoRA is $\boldsymbol{y} = p_{16}(\boldsymbol{W}_0^{\mathrm{NF4}})\boldsymbol{x} + \gamma_r \boldsymbol{B}\boldsymbol{A}\boldsymbol{x}$.

### B.3.3 Weight-Decomposed Low-Rank Adaptation (DoRA)

LoRA makes simple changes to the model weights, so it sometimes doesn't capture the full complexity of the data and its relationships. DoRA solves this issue of capturing data complexity.

DoRA decomposes the weight matrix into a *magnitude vector* and a *direction matrix*. The magnitude vector consists of the lengths of the columns in the weight matrix and is computed by taking each column's $\ell_2$ norm. The direction matrix $\boldsymbol{V}$ is the collection of the original columns. Its unit-column form $\widehat{\boldsymbol{V}} = \boldsymbol{V}/\|\boldsymbol{V}\|_c$ is obtained by dividing each column by its $\ell_2$ norm.

The magnitude vector $\boldsymbol{m}$ is of size $1 \times k$, where $k$ is the number of columns. The direction matrix $\boldsymbol{V}$ is of size $d \times k$, where $d$ is the number of rows in a weight matrix.

The decomposition can be written as:

$$\boldsymbol{W}_0 \ = \ \boldsymbol{m} \, \frac{\boldsymbol{V}}{\|\boldsymbol{V}\|_c} \ = \ \|\boldsymbol{W}_0\|_c \, \frac{\boldsymbol{W}_0}{\|\boldsymbol{W}_0\|_c},$$

where $\|\cdot\|_c$ denotes the column-wise $\ell_2$ norm (i.e., the norm is taken independently for each column) and $\boldsymbol{W}_0$ is the frozen pretrained weight.

Here is an example of the decomposition:

$$\boldsymbol{W}_0 = \begin{bmatrix} 1 & 7 & 2 & 8 & 5 \\ 2 & 10 & 4 & 12 & 10 \\ 3 & 15 & 12 & 18 & 27 \\ 4 & 12 & 16 & 16 & 36 \end{bmatrix}, \qquad \boldsymbol{W}_0 \in \mathbb{R}^{4 \times 5}.$$

For column $j$

$$\|\boldsymbol{w}_j\|_2 = \sqrt{\sum_{i=1}^{4} W_{ij}^2}.$$

These norms form a $1 \times 5$ magnitude vector:

$$\boldsymbol{m} = [5.4772, \ 22.7596, \ 20.4939, \ 28.0713, \ 46.3681].$$

The unit-column direction matrix is

$$\widehat{\boldsymbol{V}} = \begin{bmatrix} 0.182574 & 0.307562 & 0.097590 & 0.284988 & 0.107833 \\ 0.365148 & 0.439375 & 0.195180 & 0.427482 & 0.215666 \\ 0.547723 & 0.659062 & 0.585540 & 0.641223 & 0.582297 \\ 0.730297 & 0.527250 & 0.780720 & 0.569976 & 0.776396 \end{bmatrix}.$$

Every column of $\widehat{\boldsymbol{V}}$ now has unit length:

$$\|\boldsymbol{v}_j\|_2 = 1, \qquad \text{for all } j.$$

These are updated separately. The magnitude vector $\boldsymbol{m}$ is trained directly, while the direction matrix $\boldsymbol{V}$ is fine-tuned using LoRA: $\Delta \boldsymbol{V} = \boldsymbol{B}\boldsymbol{A}$ with $\boldsymbol{B} \in \mathbb{R}^{d \times r}$ and $\boldsymbol{A} \in \mathbb{R}^{r \times k}$.

After the updates, the new weight matrix is

$$\boldsymbol{W}' = \boldsymbol{m} \, \frac{\boldsymbol{V} + \Delta \boldsymbol{V}}{\|\boldsymbol{V} + \Delta \boldsymbol{V}\|_c} = \boldsymbol{m} \, \frac{\boldsymbol{W}_0 + \boldsymbol{B}\boldsymbol{A}}{\|\boldsymbol{W}_0 + \boldsymbol{B}\boldsymbol{A}\|_c}.$$

### B.3.4 Rank-Stabilized LoRA (rsLoRA)

LoRA scales the weight matrix update $\boldsymbol{B}\boldsymbol{A}$ by $\alpha/r$, which can cause gradients to explode or diminish as the rank $r$ increases. In contrast, rsLoRA uses a scaling factor $\alpha/\sqrt{r}$: $\boldsymbol{W}' = \boldsymbol{W}_0 + \frac{\alpha}{\sqrt{r}}\boldsymbol{B}\boldsymbol{A}$. This scaling results in gradient-scale stability at higher ranks, enabling the rank to be higher to capture more details in long-context tasks like XBRL extraction. rsLoRA also results in lower perplexity—the model assigns higher probabilities to correct words—than LoRA at higher ranks.

### B.3.5 LoRA with Federated Learning

In the finance sector, multiple banks may want to work together on a model to predict credit risk and whether a borrower will default on a loan. Each bank may have a different dataset, but they cannot share their data due to compliance reasons and privacy concerns. Federated learning solves this issue by fine-tuning a model on local data and aggregating updates during backpropagation to a centralized model via a server.

Differentially Private Low-Rank Adaptation (DP-LoRA) Liu et al. (2025) is a method to use federated learning with LoRA.

DP-LoRA first uses a server to send the current global LoRA weights (the A and B matrices from earlier) to all clients.

Every client does the following: 1) Gets a minibatch of its private data 2) Computes the gradient for only its local A and B weights clipped with an $\ell_2$ norm (square root of the sum of the squares of elements in the vector) 3) Adds Gaussian noise to the gradients 4) Updates the A and B LoRA matrices 5) Sends the updated A and B matrices to the server.

By adding noise, DP-LoRA prevents the centralized model from inferring the private data later on. This would allow the banks in the credit risk example to work on a model together.

As in normal federated learning, the server then aggregates the weights from all clients in a weighted average and sends the updated weights to all clients.

## C  EXPERIMENTAL SETUP

### C.1  GPUs AND APIs USED

For Llama 3.1 8B Instruct, we use four NVIDIA RTX A5000 GPUs, each with 24GB of GPU memory,, for all fine-tuning and a single A5000 for all inference. For Gemini 2.0 Flash Lite, we use Google Cloud Vertex AI for fine-tuning and inference. For the rest of the baseline models, we use the OpenAI, Together AI, DeepSeek, and Fireworks.ai API services for inference.

### C.2  FINE-TUNING

#### C.2.1  HYPERPARAMETERS AND CONFIGURATIONS

All Llama models fine-tuning experiments were conducted using the Axolotl library, using its configuration files to manage hyperparameters. The base model for all experiments was `meta-llama/Llama-3.1-8B-Instruct`. A consistent learning rate of $1 \times 10^{-4}$ was utilized across all fine-tuning runs. Table 10 outlines the common hyperparameters and settings derived from the Axolotl configuration template that were largely consistent across experiments, unless specified otherwise in the experiment-specific configurations. For Gemini 2.0 Flash Lite, we use Google Cloud Vertex AI for fine-tuning using the same number of epochs as Llama models and default settings for the rest of the hyperparameters. The Gemini 2.0 Flash Lite hyperparameters are mostly hidden by Google.

Table 10: Common Axolotl Hyperparameters and Settings.

| Parameter | Value / Setting |
| --- | --- |
| Base Model | `meta-llama/Llama-3.1-8B-Instruct` |
| Model Type | `LlamaForCausalLM` |
| Tokenizer Type | `AutoTokenizer` |
| Chat Template | `llama3` |
| Special Pad Token | `<|end_of_text|>` |
| Sequence Length | 4096 |
| Adapter (underlying mechanism) | `lora` |
| LoRA Alpha | 16 |
| LoRA Dropout | 0.05 |
| LoRA Target Modules | `q_proj, v_proj, k_proj` |
| Optimizer | `adamw_bnb_8bit` (for 8-bit quantization); `adamw_torch_fused` (for 4-bit quantization) |
| Learning Rate Scheduler | `cosine` |
| Base Learning Rate | $1 \times 10^{-4}$ |
| Warmup Steps | 10 |
| Weight Decay | 0.0 |
| Gradient Checkpointing | Enabled |
| BF16 Precision | `auto` |
| TF32 Precision | Disabled |
| Flash Attention | Disabled |
| DeepSpeed Configuration | `deepspeed_configs/zero1.json` |
| Dataset Format | Instruction-following: `context` (instruction), `target` (output). |
| Format | `[INST] {instruction} [/INST]` |
| Validation Set Size | 2% of training data |
| W&B Project Name | `finlora_models` |
| Logging Steps | Every 500 steps |
| Evaluations per Epoch | 4 |
| Checkpoints per Epoch | 1 |

Table 11 provides a detailed breakdown of the hyperparameters for each specific fine-tuning run, grouped by the PEFT method employed.

Table 11: Experiment-specific Fine-tuning Hyperparameters. All runs use `meta-llama/Llama-3.1-8B-Instruct` and a learning rate of $1 \times 10^{-4}$.

| Adapter Task | Train Dataset | Epochs | Batch size | Grad. Accum. | Effective batch size |
|---|---|---|---|---|---|
| *LoRA, 8-bit Quantization, Rank 8* | | | | | |
| Sentiment | finlora_sentiment_train | 4 | 8 | 2 | 16 |
| Headline | headline_train | 4 | 8 | 2 | 16 |
| NER | ner_train | 4 | 8 | 2 | 16 |
| Certificate | certificate_train | 4 | 8 | 2 | 16 |
| XBRL Term | xbrl_term_train | 1 | 4 | 2 | 8 |
| XBRL Tagging | finer_train_batched | 4 | 1 | 8 | 8 |
| XBRL Analysis | xbrl_extract_train | 1 | 1 | 8 | 8 |
| Formula | formula_train | 1 | 4 | 2 | 8 |
| FinanceBench | financebench_train | 4 | 1 | 2 | 2 |
| *QLoRA, 4-bit Quantization, Rank 4* | | | | | |
| Sentiment | finlora_sentiment_train | 4 | 8 | 2 | 16 |
| Headline | headline_train | 4 | 8 | 2 | 16 |
| NER | ner_train | 4 | 8 | 2 | 16 |
| Certificate | certificate_train | 4 | 8 | 2 | 16 |
| XBRL Term | xbrl_term_train | 1 | 4 | 2 | 8 |
| XBRL Tagging | finer_train_batched | 4 | 1 | 8 | 8 |
| XBRL Analysis | xbrl_extract_train | 1 | 1 | 8 | 8 |
| Formula | formula_train | 1 | 4 | 2 | 8 |
| FinanceBench | financebench_train | 4 | 1 | 2 | 2 |
| *DoRA, 8-bit Quantization, Rank 8* | | | | | |
| Sentiment | finlora_sentiment_train | 4 | 8 | 2 | 16 |
| Headline | headline_train | 4 | 8 | 2 | 16 |
| NER | ner_train | 4 | 8 | 2 | 16 |
| Certificate | certificate_train | 4 | 8 | 2 | 16 |
| XBRL Term | xbrl_term_train | 1 | 4 | 2 | 8 |
| XBRL Tagging | finer_train_batched | 4 | 1 | 8 | 8 |
| XBRL Analysis | xbrl_extract_train | 1 | 1 | 8 | 8 |
| Formula | formula_train | 1 | 4 | 2 | 8 |
| FinanceBench | financebench_train | 4 | 1 | 2 | 2 |
| *rsLoRA, 8-bit Quantization, Rank 8* | | | | | |
| Sentiment | finlora_sentiment_train | 4 | 8 | 2 | 16 |
| Headline | headline_train | 4 | 8 | 2 | 16 |
| NER | ner_train | 4 | 8 | 2 | 16 |
| Certificate | certificate_train | 4 | 8 | 2 | 16 |
| XBRL Term | xbrl_term_train | 1 | 4 | 2 | 8 |
| XBRL Tagging | finer_train_batched | 4 | 1 | 8 | 8 |
| XBRL Analysis | xbrl_extract_train_batched | 1 | 1 | 8 | 8 |
| Formula | formula_train | 1 | 4 | 2 | 8 |
| FinanceBench | financebench_train | 4 | 1 | 2 | 2 |

## C.3 INFERENCE

We use temperature 0.0 for all inference. For all fine-tuned Llama models, we use 8-bit quantization for inference. For all inference, the maximum new token lengths for inference for various financial datasets are: 10 tokens for FPB, FiQA SA, TFNS, NWGI, NER, and Headline; 20 tokens for Tag Extraction and Value Extraction; 30 tokens for Formula Construction and Formula Calculation; 50 tokens for XBRL Term, FinanceBench, and Financial Math; and 100 tokens for FiNER and FNXL.

## D   XBRL TAGGING/ANALYSIS DEMO

To help readers understand XBRL tasks and how our fine-tuned models' performance compares with base models' performance, we have created an interactive demo showcasing two tasks: XBRL tagging and XBRL analysis. The user can select one of the built-in examples as input or enter their custom prompt.

### D.1   XBRL TAGGING DEMO

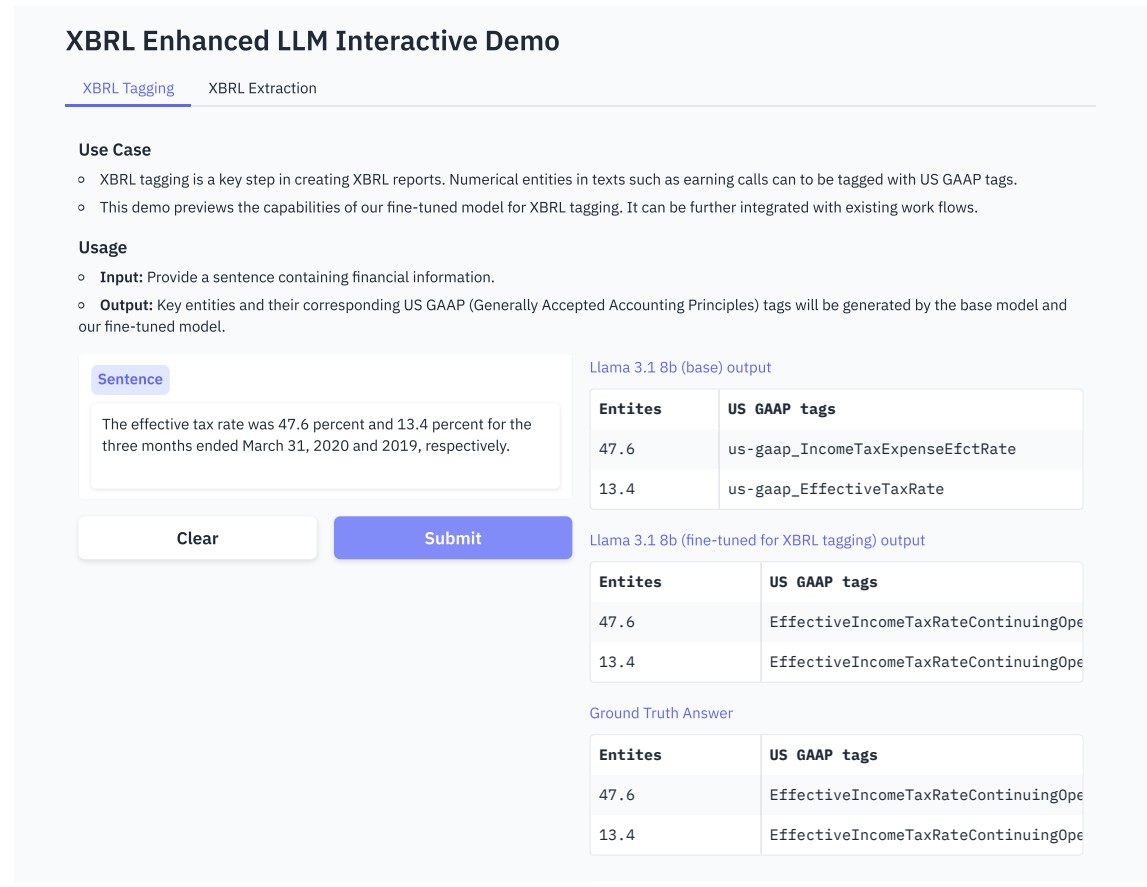

Figure 5: XBRL tagging demo.

The XBRL tagging demo showcases base and fine-tuned models designed to identify and tag numerical entities within financial texts, such as earning calls, with appropriate US GAAP (Generally Accepted Accounting Principles) taxonomies—a critical step in generating XBRL reports. Users can input a sentence containing financial data, and the system will output the key entities and their corresponding US GAAP tags as identified by both a base model and the enhanced fine-tuned model, illustrating the latter's advanced capabilities and potential for integration into existing workflows.

### D.2 XBRL ANALYSIS DEMO

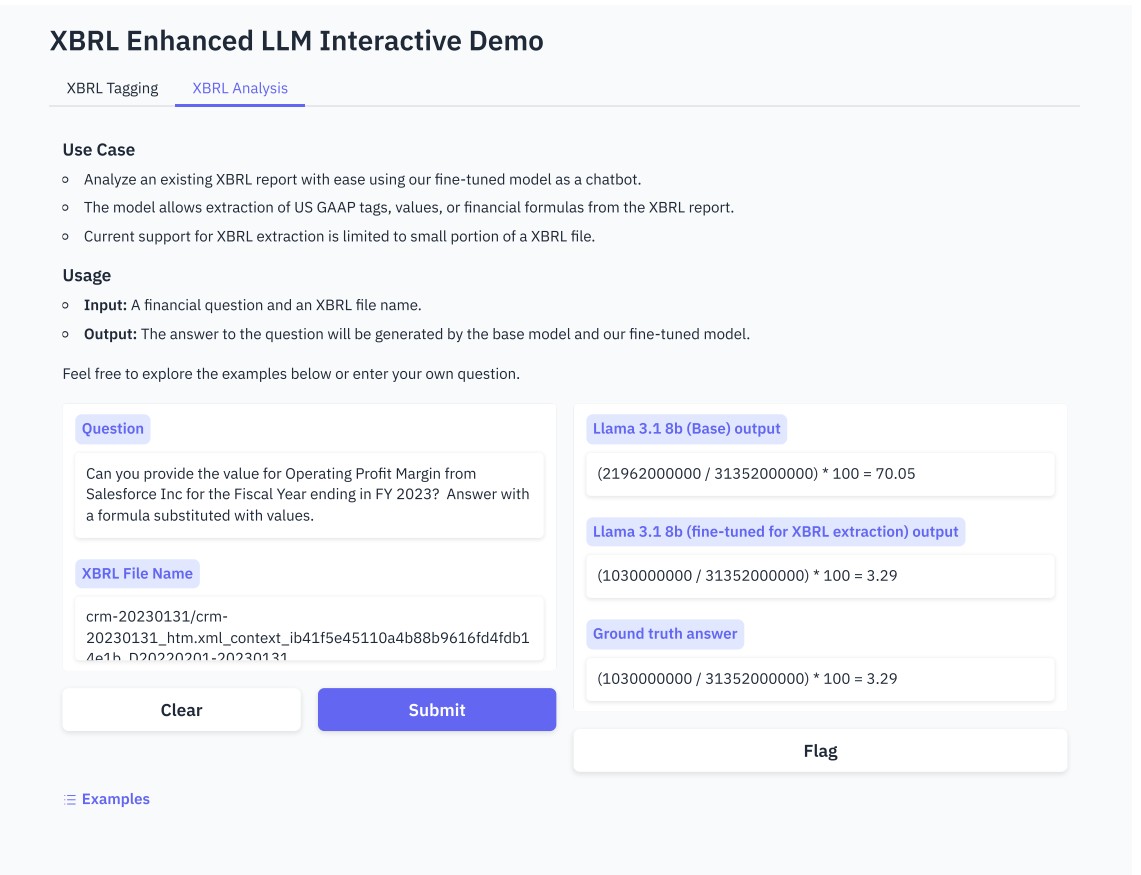

Figure 6: XBRL analysis demo.

The XBRL analysis demo showcases base and fine-tuned models acting as chatbots to simplify the analysis of existing XBRL reports. It enables users to extract US GAAP tags, values, or financial formulas, although current support is limited to analyzing small portions of an XBRL file. To use it, a user inputs a financial question along with an XBRL file name, and the system provides answers generated by both a base model and the fine-tuned model.

## E    DATASHEET FOR XBRL ANALYSIS DATASETS

The following section answers the questions listed in Datasheets for Datasets Gebru et al. (2021) for our newly constructed XBRL datasets.

### E.1    MOTIVATION

- **For what purpose were the datasets created?**
  XBRL is a key format in financial report, but XBRL data analysis is rarely explored in LLM benchmark.

- **Who funded the creation of the datasets? If there is an associated grant, please provide the name of the grantor and the grant name and number.**
  N/A.

### E.2    COMPOSITION

- **What do the instances that comprise the datasets represent (e.g., documents, photos, people, countries)?**
  The datasets represent questions and answers derived from XBRL reports.

- **How many instances are there in total (of each type, if appropriate)?**
  In total, there are about 34.2k instances in the new XBRL datasets. There are around 13k instances of Tags Extraction (10.1k train; 2.9k test), 12.6k instances of Values Extraction (10.1k train; 2.5k test), 4.3k instances of Formula Construction (3.4k train; 835 test), and 4.3k instances of Formula Calculation (3.4k train; 835 test).

- **Do the datasets contain all possible instances or is it a sample (not necessarily random) of instances from a larger set?**
  The datasets contain all instances of the questions we curated.

- **What data does each instance consist of?**
  - instruction: The instruction for the LLM to work on this task.
  - input: The question.
  - output: The ground truth answer.
  - year: The year of the XBRL report the extract is based on.
  - company: The ticker symbol of the company the XBRL report belongs to.
  - doc_path: The relative path of the XBRL report required for this entry. Zipped XBRL reports are within the Hugging Face dataset repository in Git LFS format.
  - context_id: In XBRL, the `context_id` is the unique identifier for a `<context>` element, which defines crucial metadata for reported financial facts, such as the reporting period, and any applicable dimensional information (like business segments or scenarios). We provide the relevant `context_id` for each entry in this dataset to narrow down the scope of data the LLM needs to consider. This pre-selection of the context ensures that the LLM can focus on the question using a targeted set of facts, rather than search the entire XBRL report. Finding the appropriate `context_id` is not considered part of the LLM's task for several reasons: for questions related to consolidated, company-wide figures for a specific year (which is the focus of our dataset), the correct `context_id` can generally be identified programmatically. This typically involves filtering the context definitions by the target date or year, and then selecting the context that represents the consolidated entity (often identifiable by its lack of dimensional axis information). This process is a deterministic and straightforward data preprocessing step that can be efficiently implemented with code, making it unnecessary to leverage an LLM for this aspect of data retrieval.

- **Is there a label or target associated with each instance?**
  Yes, each question in the dataset has an associated ground truth answer, referred to as "output".

- **Is any information missing from individual instances? If so, please describe why.**
  No.

- **Are relationships between individual instances made explicit (e.g., users' movie ratings, social-network links)?**
  N/A. Each instance is generally treated as independent.

- **Are there recommended data splits (e.g., training, development/validation, testing)?**
  Yes, the data is split chronologically by year. Questions based on XBRL reports from 2019-2022 form the training split, while questions based on 2023 XBRL reports constitute the test split. This approach is used to ensure no data leakage, as XBRL reports often contain comparative data from previous years.

- **Are there any errors, sources of noise, or redundancies in the dataset?**
  To the best of our knowledge, no.

- **Are the datasets self-contained, or do they link to or otherwise rely on external resources?**
  The dataset entries do not directly embed the full raw XBRL text, even when filtered by `context_id`, due to the considerable size of these texts. However, we provide code in our GitHub repository that allows users to fetch the source XBRL documents and reconstruct the complete data for each entry. Otherwise, the structured data provided is self-contained.

- **Do the datasets contain data that might be considered confidential?**
  No.

- **Do the datasets contain data that, if viewed directly, might be offensive, insulting, threatening, or might otherwise cause anxiety?**
  No.

- **Do the datasets relate to people?**
  No.

- **Do the datasets identify any sub-populations (e.g., by age, gender)?**
  No.

- **Is it possible to identify individuals, directly or indirectly, from the datasets?**
  No.

- **Do the datasets contain data that might be considered sensitive in any way?**
  No.

## E.3 COLLECTION PROCESS

- **How was the data associated with each instance acquired? Was the data directly observable (e.g., raw text, movie ratings), reported by subjects (e.g., survey responses), or indirectly inferred/derived from other data (e.g., part-of-speech tags, model-based guesses for age or language)? If data was reported by subjects or indirectly inferred/derived from other data, was the data validated/verified? If so, please describe how.**
  The questions were generated based on 22 financial concepts and 7 financial formulas. The ground truth answer were extracted from the XBRL documents using the Arelle package.

- **What mechanisms or procedures were used to collect the data? How were these mechanisms or procedures validated?**
  N/A.

- **If the datasets are a sample from a larger set, what was the sampling strategy?**
  No. The provided datasets are complete.

- **Who was involved in the data-collection process and how were they compensated?**
  N/A.

- **Over what timeframe was the data collected? Does this timeframe match the creation timeframe of the data?**
  N/A.

- **Were any ethical-review processes conducted (e.g., by an institutional review board)?**
  No.

- **Do the datasets relate to people?**
  No.

- **Did you collect the data directly from the individuals in question, or obtain it via third parties or other sources?**
  N/A.

- **Were the individuals in question notified about the data collection?**
  N/A.

- **Did the individuals in question consent to the collection and use of their data?**
  N/A.

- **If consent was obtained, were the consenting individuals provided with a mechanism to revoke their consent in the future or for certain uses?**
  N/A.

- **Has an analysis of the potential impact of the dataset and its use on data subjects (e.g., a data-protection impact analysis) been conducted?**
  No.

### E.4 PREPROCESSING / CLEANING / LABELING

- **Was any preprocessing, cleaning, or labeling of the data done?**
  The Arelle package was used to process XBRL files in order to create the ground truth answer.

- **Was the raw data saved in addition to the preprocessed/cleaned/labeled data?**
  Raw XBRL files will be available for download at Hugging Face.

### E.5 USES

- **Have the datasets been used for any tasks already?**
  The dataset is used for XBRL analysis tasks.

- **Is there a repository that links to any or all papers or systems that use the datasets?**
  N/A.

- **What other tasks could the datasets be used for?**
  XBRL/financial statement analysis related tasks.

- **Is there anything about the datasets' composition or collection that might impact future uses?**
  No.

- **Are there tasks for which the datasets should *not* be used?**
  N/A.

### E.6 DISTRIBUTION

- **Will the datasets be distributed to third parties outside the entity on behalf of which the dataset was created?**
  Dataset will be released publicly after review period ends.

- **When will the datasets be distributed?**
  The datasets are available now.

- **Will the datasets be distributed under a copyright or other intellectual property (IP) license, and/or under applicable terms of use (ToU)?**
  The datasets are released under the MIT License.

- **Have any third parties imposed IP-based or other restrictions on the data?**
  No.

- **Do any export controls or other regulatory restrictions apply to the datasets or to individual instances?**
  No.

### E.7    MAINTENANCE

- **Who will support, host, or maintain the datasets?**
  We, the authors of the FinLoRA paper, will.

- **Is there an erratum?**
  No.

- **If the datasets relate to people, are there applicable limits on the retention of the data associated with the instances (e.g., were the individuals in question told that their data would be retained for a fixed period of time and then deleted)?**
  N/A.

- **If others want to extend/augment/build on/contribute to the dataset, is there a mechanism for them to do so?**
  Yes, using Open Review Comments.

