# OpenReview forum: "FinLoRA: Benchmarking LoRA Methods for Fine-tuning LLMs on Financial Datasets"
_ICLR.cc/2026/Conference — Submitted to ICLR 2026_

### Official Review · Reviewer_JkuD · 2025-10-18

**Soundness:** 3
**Presentation:** 3
**Contribution:** 3
**Rating:** 4
**Confidence:** 4

**Summary:**

This paper proposes FinLORA, aiming for benchmarking current large language models (LLMs) abilities on solving real-world financial tasks. Specifically, authors combine existing popular financial application tasks and introduce novel datasets. Experimental results across five LoRA methods and diverse base LLMs provide several insights, which may be meaningful for the relevant domain.

**Strengths:**

- Good presentation, easy to read and follow.
- The research topic is interesting and timely.

**Weaknesses:**

1. The paper evaluates only 5 LoRA methods (LoRA, QLoRA, DoRA, rsLoRA, Federated LoRA) but omits several important variants. Missing methods include:
- AdaLoRA [1], which adaptively allocates rank budget
- LoRA+ [2],  which uses different learning rates for A and B matrices
- VeRA [3], which shares LoRA matrices across layers
Considering the main benchmarking of this paper is LoRA and variants, it is necessary to include at least 2-3 additional LoRA variants to make the benchmark more comprehensive and useful for future researchers.

2. Table 8 shows task similarity correlates with multi-task performance, but the analysis lacks depth:
- Why does Financial Statement Analysis benefit (+7.31 points) while General Financial suffers (-27.22 points)?
- No investigation of task-specific vs. shared knowledge.

3. The paper acknowledges rsLoRA benefits appear "primarily at high ranks" (page 6, lines 292-297) but only tests r=8. Original rsLoRA paper demonstrates benefits at r=64, 128. Thus, the paper's conclusion that rsLoRA "just slightly underperforms" is misleading given the experimental setup.

4. The paper compares against general-purpose LLMs (GPT-4o, Llama 70B) but misses comparison with other financial LLMs: FinBERT, FinGPT. The performance gap is important for analysis.

5. The novel XBRL datasets are a key contribution, but construction details are insufficient (page 3, lines 132-143):

- "Five distinct templates" mentioned but not shown or analyzed for diversity.
- No discussion of template design process or validation.

6. The inference analysis (Section 4.3, Figure 3) is incomplete: only wall-clock time measured, not throughput (tokens/second) and
no analysis of different batch sizes (only mentions larger batches could help).



[1]https://arxiv.org/abs/2303.10512

[2]https://arxiv.org/abs/2402.12354

[3]https://arxiv.org/abs/2310.11454

**Questions:**

1. In Table 2, what does the 'Average prompt length' mean? token length or string length?

2. The meaning of caption of Figure 2 is vague. Please consider revision.

3. What is the difference between single-task and multi-task fine-tuning? Did you control the same number of samples utilized? More details are needed.

4. What is the performance gap between LoRA variants and full fine-tuning?

5. Regarding Federated LoRA, more details of this are needed. (No citations in the main paper or appendix)

---

### Official Review · Reviewer_SWPW · 2025-11-01

**Soundness:** 2
**Presentation:** 2
**Contribution:** 2
**Rating:** 2
**Confidence:** 4

**Summary:**

This paper presents FinLoRA, a benchmark for evaluating Low-Rank Adaptation (LoRA) methods on financial tasks. The authors curate 19 datasets spanning general financial tasks, financial certificate exams, financial reporting, and financial statement analysis. Notably, they introduce 4 novel XBRL (eXtensible Business Reporting Language) analysis datasets constructed from SEC 10-K filings of Dow Jones 30 companies (2019-2023). The benchmark evaluates five LoRA variants—vanilla LoRA, QLoRA, DoRA, rsLoRA, and Federated LoRA—across multiple base models (Llama 3.1 8B/70B, Gemini 2.0 Flash Lite, DeepSeek V3, GPT-4o). Results show LoRA fine-tuning achieves an average 40.1-point accuracy improvement over base models, with vanilla LoRA (8-bit, rank 8) achieving the highest overall score (74.74). The paper also analyzes computational costs, finding local LoRA fine-tuning costs $14.66-$16.54 versus $162-$312 for cloud API services, and investigates practical deployment considerations including federated learning and catastrophic forgetting.

**Strengths:**

**Novel XBRL datasets address a real gap**: The four new XBRL analysis datasets (tag extraction, value extraction, formula construction, formula calculation) fill a genuine void in financial NLP benchmarking. XBRL is the de facto standard for SEC filings, yet dedicated datasets for XBRL analysis tasks are scarce. The construction methodology using filtered XBRL segments with context IDs is sound, and the datasets could enable future research in automated financial report analysis.

**Weaknesses:**

**Misuse of the term “benchmark.”** The paper calls itself a benchmark but only evaluates LoRA fine-tuning variants. A real benchmark must be method-agnostic—supporting any model type (base, instruction-tuned, full fine-tuned, PEFT variants, and proprietary frontier models). This is not a benchmark but a *LoRA comparison study*. You benchmark *tasks*, not *methods*. True benchmarks like MMLU, GSM8K, HumanEval, or GPQA allow any model to be tested. Restricting evaluation to LoRA makes this unusable for broader comparison.

**Lack of frontier models.** The study includes GPT-4o and DeepSeek V3 but ignores current top models (GPT-4.1, GPT-5, Claude 4.1/Opus 4, Gemini 2.0 Pro, etc.). Without assessing SOTA models, we can’t tell whether the tasks remain challenging or already saturated—undermining the benchmark’s relevance.

**Incomplete open-source coverage.** Important open-source baselines (Qwen 3, Kimi-Chat, etc.) are missing. Testing only Llama 3.1 and Ministral offers no evidence that findings generalize across architectures or model families.

**Unproven task difficulty.** The benchmark’s usefulness depends on whether it still challenges top models. Yet the paper reports only LoRA and base Llama results (74.74% vs 66.44%) without showing upper bounds from frontier systems. If GPT-4.1 or Claude Opus already reach 95%+, the benchmark is instantly obsolete.

**Questions:**

1. **Why is this called a "benchmark" when it only evaluates LoRA methods?** A benchmark should be method-agnostic. Why not evaluate full fine-tuning, other PEFT methods (prefix tuning, adapters, prompt tuning), instruction tuning, and few-shot prompting? How can researchers use your "benchmark" if they develop a new fine-tuning method that isn't LoRA-based?

2. **What is the performance of frontier models on these tasks?** Please report results for GPT-4.1, GPT-5 (if available), Claude 4.1 Sonnet, Claude Opus 4, Gemini 2.0 Pro, and other current SOTA models. If these models achieve >90% accuracy, does your benchmark still provide value for measuring future progress? A benchmark must demonstrate it is challenging for the best available models.

---

### Official Review · Reviewer_B1F8 · 2025-11-01

**Soundness:** 3
**Presentation:** 3
**Contribution:** 3
**Rating:** 4
**Confidence:** 3

**Summary:**

This paper introduces FinLoRA, a benchmark designed to evaluate various Low-Rank Adaptation (LoRA) methods for fine-tuning Large Language Models (LLMs) on financial tasks. The authors curate 19 datasets, including four novel ones for XBRL (eXtensible Business Reporting Language) analysis derived from SEC filings. They systematically benchmark five LoRA variants (LoRA, QLoRA, DoRA, rsLoRA, Federated LoRA) across five base LLMs (e.g., Llama 3.1 8B, Gemini 2.0 Flash) on tasks ranging from sentiment analysis to professional financial certification and complex XBRL formula manipulation. The central claims are: (i) LoRA methods provide substantial performance gains (e.g., +40.1 points accuracy for vanilla LoRA) over base models in finance; (ii) The effectiveness of LoRA varies significantly across financial task types, with structured data like XBRL being particularly amenable; (iii) LoRA offers a highly cost-effective alternative to full fine-tuning; and (iv) LoRA, at low ranks, does not lead to catastrophic forgetting. The paper positions FinLoRA as a resource to democratize financial AI.

**Strengths:**

Comprehensive Benchmarking: The scale of the evaluation is a significant strength, encompassing 19 datasets, 5 base models, and 5 LoRA methods. This provides a rich, multi-faceted view of the landscape.

Novel and Valuable Datasets: The introduction of four XBRL analysis datasets fills a genuine gap in the literature. The tasks (tag/value extraction, formula construction/calculation) are well-chosen to test sophisticated financial reasoning and will be a valuable resource for future research.

Practical Relevance: The paper thoroughly addresses practical concerns like computational cost, inference latency, and deployment considerations (federated learning, catastrophic forgetting), which are critical for real-world adoption in the finance industry.

Clarity of High-Level Findings: The paper successfully communicates its main messages: the dramatic performance gains from LoRA, the cost-effectiveness, the variation in task suitability, and the absence of catastrophic forgetting at low ranks.

**Weaknesses:**

Lack of Statistical Rigor: As noted, the most significant weakness is the presentation of results without any measure of variance or statistical significance. This is a critical omission for a benchmark paper claiming to compare methods.

Insufficient Hyperparameter Investigation: The comparison of LoRA variants is not equitable because it does not optimize hyperparameters for each method. The fixed, low-rank (r=8) setup particularly disadvantages rsLoRA, which is designed for higher ranks. The explanation for DoRA's underperformance is speculative and should have been validated with experiments using separate learning rates.

Superficial Analysis of Key Results:

The intriguing result on "negative transfer" in multi-task learning is mentioned but not deeply analyzed. A discussion or hypothesis on why certain tasks conflict would significantly strengthen the paper.

The analysis of why XBRL tasks benefit more from LoRA than FinanceBench is good but could be taken further. A quantitative analysis of the data quality (e.g., token-level consistency, noise levels) could provide more concrete evidence.

Reproducibility Concerns: The omission of key experimental details (optimizer, prompts, alpha value, federated learning parameters) and the incomplete references hinder the reproducibility of this work.

**Questions:**

1. Statistical Significance: Can the authors report the mean and standard deviation of the primary metrics (e.g., accuracy) over at least 3 independent training runs for a subset of the key experiments (e.g., comparing LoRA variants on Llama 3.1 8B)? This is essential to confirm that the performance differences you highlight are statistically significant.

2. Hyperparameter Sensitivity: Given that the performance of DoRA and rsLoRA is known to be sensitive to their specific configurations, can you provide results from a hyperparameter sweep (e.g., over rank r and learning rates for the magnitude/direction components in DoRA)? This would ensure a fair comparison and validate your hypotheses about their potential.

3. Federated Learning Details: Could you provide a detailed description of your federated learning simulation, including the number of federation rounds, the number of local epochs per round, whether the data was partitioned in an IID or non-IID manner, and the specific aggregation algorithm used? This is crucial for interpreting the federated LoRA results.

4. Reproducibility: Please provide the full set of hyperparameters (optimizer, LoRA alpha, dropout, prompt templates) used for fine-tuning each model, and ensure all citations are complete with full bibliographic entries, especially for preprints.

---

### Official Review · Reviewer_uUoz · 2025-11-01

**Soundness:** 1
**Presentation:** 2
**Contribution:** 1
**Rating:** 2
**Confidence:** 4

**Summary:**

This paper introduces a benchmark project that is curated with 19 datasets, covering four types of financial tasks: general financial tasks, financial certificate tasks, financial reporting tasks and financial statement analysis tasks. 15 datasets are from existing public archives, and the other 4 are constructed and contributed originally. This paper then implements five LoRA methods on the datasets: Vanilla LoRA, QLoRA, DoRA, rsLoRA and Federated LoRA,  comparing their performance metrics, task suitability, resource consumption and discussing practical considerations for their real-world deployment.

**Strengths:**

1. This paper contributed four new datasets for XBRL analysis tasks, which serve as a supplement to current data resources.
2. Experiment examines five LoRA variants on 19 datasets, which contains lots of observations.
3. This paper has identified four angles to analyze LoRA methods in financial fine-tuning tasks. The conclusion that LoRA does not exhibit catastrophic forgetting after being fine-tuned on these datasets is interesting.

**Weaknesses:**

1. As a benchmark paper, it doesn’t contain novel algorithms. However, only 4 out of 19 datasets are newly curated as substantial contribution. Only 4 different algorithms are evaluated, which may be not enough for a nice benchmark paper.
2. Fine-tuning are majorly conducted on Llama-3.1-8B, which is limited for generalization. The comparison conclusions may not be reliable. Using FL only with Gemini for comparison is possibly unfair.
3. Averaging accuracy and BERTScore F1 is strange. Error bars are not reported.
4. For some datasets, F1-score is highly similar to Accuracy, for example, those financial certificate tasks. This might indicate some underlying problem within data set construction or some error in implementation, which needs further justification.
5. The experiment conclusions seem to be trivial within the literature. It is an easy fact that different LoRA methods can behave differently on different tasks and datasets. The costs and inference time of different LoRA methods are the properties of the method and base models, therefore reports on these terms contain no new knowledge.
6. Task suitability evaluation is also doubtful, since there are other variables that may influence the results (e.g., the quality of prompt, since prompts are different for different tasks.
7. For Catastrophic Forgetting part, it would be more informative to show whether existing continuous learning methods can still outperform simple LoRA methods or not, as supporting result.

**Questions:**

What are the key points that make the performances of these LoRA methods differentiate from each other? How this key points are related to the properties of datasets?

---

### Meta-Review · Area_Chair_PWx4 · 2026-01-06

**Summary:**

In the absence of a rebuttal, all reviewer scores remain negative, and the concerns raised during the initial review period remain unresolved.

**Reviewer Concerns:**

No rebuttal provided

**Reviewer Scores:**

No rebuttal provided, the reviewer will not change their score

---

### Decision · Program_Chairs · 2026-01-26

Reject